# Acute depletion of BRG1 reveals its primary function as an activator of transcription

Gang Ren[1,2,6], Wai Lim Ku [1,6], Guangzhe Ge[1,6], Jackson A. Hoffman [3], Jee Youn Kang[1], Qingsong Tang[1], Kairong Cui[1], Yong He[4], Yukun Guan [4], Bin Gao [4], Chengyu Liu[5], Trevor K. Archer [3] & Keji Zhao [1] ✉

The mammalian SWI/SNF-like BAF complexes play critical roles during animal development and pathological conditions. Previous gene deletion studies and characterization of human gene mutations implicate that the complexes both repress and activate a large number of genes. However, the direct function of the complexes in cells remains largely unclear due to the relatively long-term nature of gene deletion or natural mutation. Here we generate a mouse line by knocking in the auxin-inducible degron tag (AID) to the *Smarca4* gene, which encodes BRG1, the essential ATPase subunit of the BAF complexes. We show that the tagged BRG1 can be efficiently depleted by osTIR1 expression and auxin treatment for 6 to 10 h in CD4 + T cells, hepatocytes, and fibroblasts isolated from the knock-in mice. The acute depletion of BRG1 leads to decreases in nascent RNAs and RNA polymerase II binding at a large number of genes, which are positively correlated with the loss of BRG1. Further, these changes are correlated with diminished accessibility at DNase I Hypersensitive Sites (DHSs) and p300 binding. The acute BRG1 depletion results in three major patterns of nucleosome shifts leading to narrower nucleosome spacing surrounding transcription factor motifs and at enhancers and transcription start sites (TSSs), which are correlated with loss of BRG1, decreased chromatin accessibility and decreased nascent RNAs. Acute depletion of BRG1 severely compromises the Trichostatin A (TSA)-induced histone acetylation, suggesting a substantial interplay between the chromatin remodeling activity of BRG1 and histone acetylation. Our data suggest BRG1 mainly plays a direct positive role in chromatin accessibility, RNAPII binding, and nascent RNA production by regulating nucleosome positioning and facilitating transcription factor binding to their target sites.

The mammalian SWI/SNF-like BAF complexes have nucleosome remodeling activities in vitro and are important regulators of animal development and pathological conditions[1–6]. They are critical for nucleosome organization to determine chromatin states in mammalian cells[4,5,7,8]. Although the BAF complexes are implicated in facilitating enhancer activities for gene activation, they are associated with silencing a large number of genes by gene deletion studies and characterization of human gene

---

[1]Systems Biology Center, National Heart, Lung, and Blood Institute, NIH, Bethesda, MD, USA. [2]College of Animal Science and Technology, Northwest Agriculture and Forest University, Yangling, Xianyang, Shaanxi, China. [3]Epigenetics and Stem Cell Biology Laboratory, National Institute of Environmental Health Sciences, NIH, Research Triangle Park, Durham, North Carolina, USA. [4]Laboratory of Liver Diseases, National Institute on Alcohol Abuse and Alcoholism, National Institutes of Health, Bethesda, MD, USA. [5]Transgenic Core Facility, National Heart, Lung, and Blood Institute, NIH, Bethesda, MD, USA. [6]These authors contributed equally: Gang Ren, Wai Lim Ku, Guangzhe Ge. ✉e-mail: zhaok@nhlbi.nih.gov

mutations[5,8–13]. BRG1, the key ATPase subunit of BAF complexes, has been proposed to function in gene regulation by creating nucleosome free regions (NFRs) via nucleosome ejection and/or sliding[4,14–20]. Recent studies demonstrated that BRG1 mainly represses transcription from promoters associated with H3K4me3 and narrow NFRs while it counteracts transcriptional repression of promoters associated with H3K4me3 and H3K27me bivalent modifications[10,21]. The function of BRG1 in transcription has been extensively investigated in several studies using various developmental and disease systems. These studies found that about half of the BRG1 target genes are repressed by BRG1 and the other half are activated by BRG1 (Supplementary Table 1[13,15,21–26]), suggesting that BRG1 functions both as a repressor and activator of transcription in cells. However, BRG1 function is traditionally studied in cell lines or animal models by constitutive or conditional gene deletion strategies, which require a time of at least several days for efficient deletion of the gene. Indirect effects may accumulate to conceal the direct function of BRG1 during several days of rapid cell proliferation or animal development.

In this work, we generate a mouse line by knocking in the auxin-inducible degron tag (AID) into the *Brg1* gene. We show that the tagged BRG1 displays a similar binding pattern as the endogenous BRG and can be efficiently depleted by osTIR1 expression and auxin treatment within 6 to 10 h in CD4 + T cells, hepatocytes, and fibroblasts isolated from these knock-in mice. The acute depletion of BRG1 reduces RNA polymerase II binding and nascent transcription, accompanied by globally decreased accessibility at DNase I Hypersensitive Sites (DHSs). The BRG1 depletion compromises the nucleosome phasing around the binding sites of key transcription factors including Gata3, T-bet, Stat5a, Ets1, and Fli1 but not CTCF, leading to narrower nucleosome spacing at enhancers and TSSs. Our data indicate that BRG1 primarily contributes positively to chromatin accessibility, RNAP binding, and nascent RNA production.

## Results

### Acute depletion of BRG1 results in loss of BRG1 binding to chromatin

To better investigate the direct target genes regulated by BRG1 in murine cells, we knocked in the auxin-induced degron (AID) to the *Brg1* gene locus, using the CRISPR-Cas9 strategy, to express the BRG1-AID fusion protein (Supplementary Fig. 1a, b). No developmental defects were observed in either heterozygous or homozygous BRG1-AID mice, which is consistent with the observation that knocking in AID did not change BRG1 protein level in cells (Fig. 1a). Further, Chromatin Immunocleavage sequencing (ChIC-seq)[27] analysis indicated that the BRG1-AID protein and wild type BRG1 exhibited highly similar binding profiles on chromatin in CD4 + T cells (Supplementary Fig. 1c, d). Thus, we used BRG1 to represent BRG1-AID hereafter. To deplete the BRG1 protein, CD4 + T cells were isolated from the lymph nodes of the AID knock-in mice, transduced with osTIR1 expressing viral particles, and treated with auxin. BRG1 was efficiently depleted after treatment with auxin for 6, 10, and 24 h (Fig. 1a; "Methods"). It is worth noting that the BRG1-AID depletion was innocuous to cell viability and growth in our experimental timescale. ChIC-seq analysis revealed that the auxin-induced depletion of BRG1 apparently reduced its binding to the vast majority of its target sites on chromatin (Fig. 1b, c). Globally, BRG1 peaks were detected at 14,000 TSSs and 16,000 at putative enhancers and other regions in CD4 + T cells (Fig. 1d). Among the 4555 significantly decreased BRG1 binding sites, 62% are at TSSs while 33% of them are at enhancers (Fig. 1e). We conclude that the AID-tagged BRG1 can be rapidly depleted upon auxin treatment in the presence of osTIR1, which is accompanied by a substantial loss of BRG1 binding at promoters and enhancers on chromatin.

### Acute depletion of BRG1 results in genome-wide decrease in nascent transcription

BRG1 is known to function as either an activator or repressor of transcription in various cells based on knockout or knockdown studies[2,4,13,18,21] (Supplementary Table 1). However, analysis of BRG1 binding at TSSs indicated a positive correlation between the BRG1 binding level and gene expression level (Fig. 2a), which suggests that BRG1 may play an active role but not repressive role in transcription. To test this directly, we performed RNA-seq analysis in T cells after the acute depletion of BRG1 for 10 h and identified 24 up-regulated and 24 down-regulated genes (Fig. 2b), which appears to support the previous observations that BRG1 acts as both an activator and repressor of gene expression. However, RNA-seq does not directly measure the transcriptional activity in cells and thus may not be able to reveal the immediate effects of the acute BRG1 depletion on transcription. By contrast, PRO-Seq measures the direct activity of RNA polymerase II engaged in chromatin[28] and can detect rapid changes of transcription activity. Thus, we applied PRO-Seq to measure nascent transcripts in BRG1-depleted and control cells. Strikingly, we found that 5127 genes were significantly down-regulated while only 93 genes were significantly up-regulated (Fig. 2c). As expected, the PRO-Seq signals were enriched in the TSS regions and BRG1 depletion resulted in the most significant decreases in 5' regions of the gene (Fig. 2d, e). To test whether the decrease in nascent RNAs is related with the loss of BRG1 upon its depletion, we found that while most genes exhibited decreased BRG1 binding after auxin treatment, the significantly down-regulated genes showed more decrease in BRG1 binding than unaffected genes (Fig. 2f). Furthermore, the gene 5' ends showed more significant decrease than the gene 3' ends (Fig. 2g), which correlated with BRG1 binding levels. Altogether, these results revealed a positive correlation between the decrease in the nascent transcription and the loss of BRG1 binding resulting from its acute depletion. These results indicate that the vast majority of BRG1-bound genes displayed apparently decreased nascent transcription upon the acute depletion of BRG1 and that BRG1 primarily functions as an activator of nascent transcription.

### Acute BRG1 depletion results in genome-wide loss of RNA Pol II binding

PRO-seq detects nascent RNAs produced by engaged RNA polymerase II bound to chromatin[28]. Thus, it is likely that acute depletion of BRG1 may result in decreased RNA polymerase II binding on chromatin, which then leads to decreased nascent transcription. To this end, we examined the binding profiles of RNA polymerase II using ChIC-seq. Our data revealed a global decrease in RNA polymerase II binding upon the acute depletion of BRG1 (Supplementary Fig. 2 and Fig. 2h). More decreases in RNA polymerase II binding were observed at the 5' ends of the gene than at the 3' ends (Fig. 2i), consistent with the changes of nascent transcripts upon depletion of BRG1. These results indicate that BRG1 plays a critical role in the recruitment of RNA polymerase II to chromatin and the production of nascent transcripts.

### Acute BRG1 depletion results in genome-wide decrease in chromatin accessibility

RNA polymerase II binding is regulated by regulatory elements such as promoters and enhancers, whose accessibility is critical for activity. To test whether BRG1 facilitates Pol II binding to chromatin by regulating the accessibility of these transcriptional regulatory elements, we measured the accessibility of chromatin in BRG1-depleted and control cells using DNase-seq[29,30]. Analysis of the data revealed significant decreases in accessibility measured by DNase-seq tag density in T cells depleted of BRG1 for 10 h as compared to control cells (Fig. 3a–c and Supplementary Fig. 3). We observed that among the 8305 (22% of all DHSs) significantly changed DHSs, 7987 sites (96.0%) showed significantly decreased accessibility after acute depletion of BRG1 while only 318 sites (4.0%) showed increased accessibility (Fig. 3d). Since the

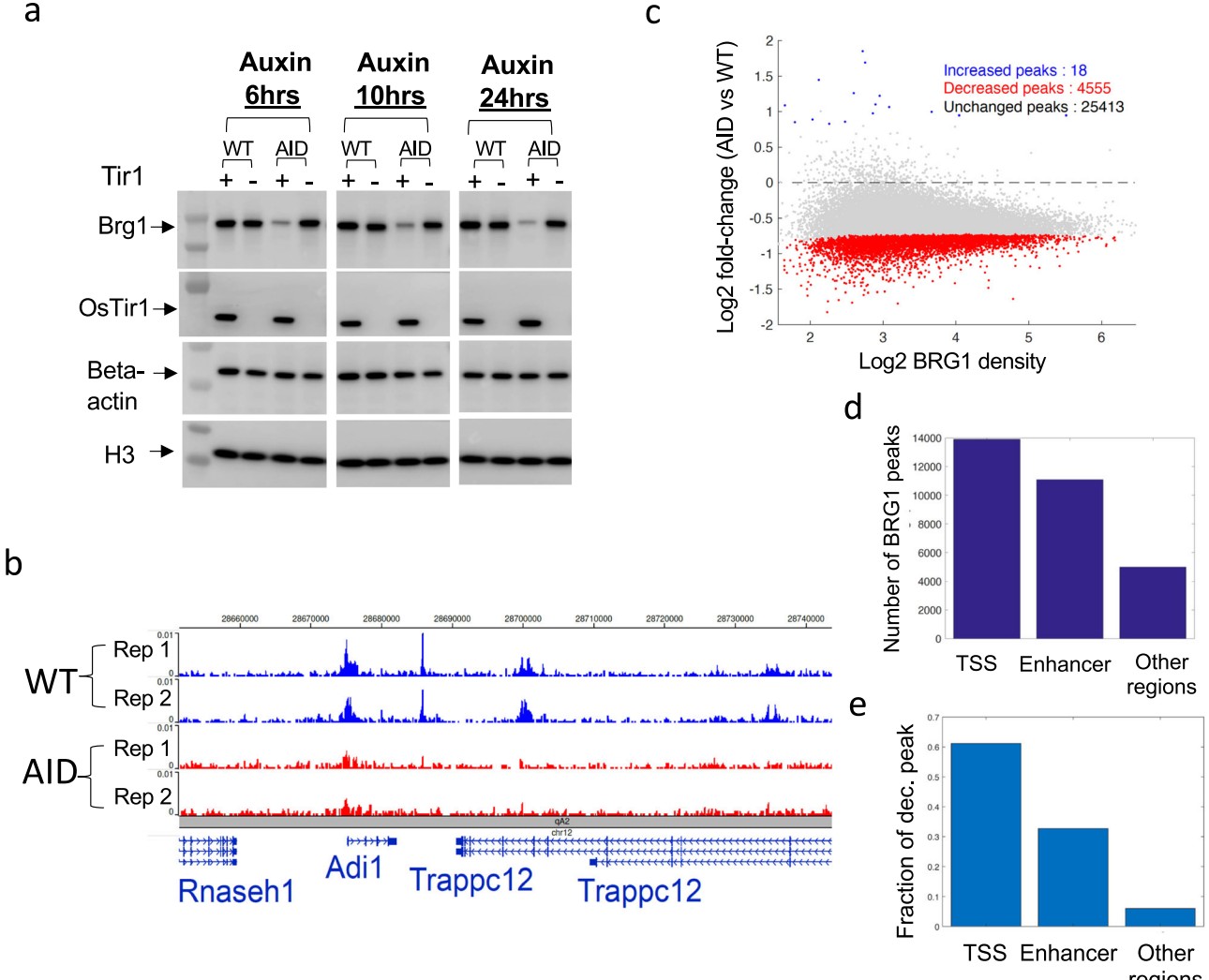

**Fig. 1 | Acute depletion of BRG1 with auxin-inducible degron (AID) system in mouse T cells results in significant decreases in BRG1 binding genome-wide.**
**a** Western blotting shows efficient depletion of BRG1 by Auxin treatment for 6, 10, and 24 h. Primary CD4 + T cells were isolated from BRG1-AID mice or wild-type mice. Following transduction of the cells with osTIR1 expression viral particles for 24 h, the cells were treated with Auxin for the times indicated on the top (6, 10, and 24 h). WT indicates the cells isolated from wildtype B6 mice; AID indicates the cells isolated from Brg1-AID knock-in B6 mice. Beta-actin and histone H3 were used as protein loading controls (repeated ≥ 2 times independently with similar results; uncropped scans of Fig. 1a are supplied as Source Data files). **b** A genome browser snapshot showing the BRG1 ChIC-seq signals in wild-type cells and BRG1-depleted cells. The cells were treated with Auxin for 10 h as described in Panel (**a**). Blue

tracks: two replicates for wild-type cells; Red tracks: two panels for BRG1-depleted cells. **c** MA plot illustrating the genome-wide reduction in BRG1 binding (data from Panel (**b**) following 10 h of BRG1 depletion. Significantly altered BRG1 peaks are indicated, with decreases in red and increases in blue, from a total of 29,986 peaks. Significance was determined using a two-sided ROTS statistical test, with a cutoff of $P < 0.05$ and fold-change > 1.3. The P-values were calculated using the two-sided ROTS method, optimizing the reproducibility of feature ranking. **d** Bar plot showing the number of the BRG1 peaks at TSS, enhancers, and other regions in the wild-type CD4 + T cells. Note that enhancers are defined by the H3K27ac ChIC-seq data. **e** Bar plot showing the fractions of decreased BRG1 peaks at TSS, enhancers, and other regions in CD4 + T cells after BRG1 depletion for 10 h.

accessibility of DHSs was positively correlated with the levels of BRG1 binding at both TSS and non-TSS regions in wild type T cells (Fig. 3e, f), we further tested whether BRG1 contributes to chromatin accessibility. We compared the decrease in DNase-seq signals and loss of BRG1 binding upon the acute depletion of BRG1 and found that the decrease in DHS accessibility was significantly correlated with the decrease in BRG1 binding (Fig. 3g, h). These results indicate that BRG1 binding promotes chromatin accessibility of transcriptional regulatory elements in the genome. In addition to the reduction in DHSs following BRG1 depletion, another noteworthy observation emerged from our analysis. We noted a corresponding decrease in H3K4me3 density at the TSS regions where BRG1 exhibited more pronounced decreases (as shown in Supplementary Fig. 4a).

## BRM014 treatment impairs chromatin accessibility and gene expression in T cells

To further confirm the dependence of chromatin accessibility on BRG1, we examined the changes of nascent transcription and DHS accessibility in CD4 + T cells after treatment with BRM014, which is a dual BRM and BRG1 inhibitor, for a duration of 10 h. As shown in the Genome Browser images, the BRM014 treatment led to substantial reductions in PRO-Seq signals (Fig. 4a). MA plots revealed that 1223 genes displayed a significant decrease in PRO-Seq signals upon BRM014 treatment, while only 35 genes exhibited augmented signals (Fig. 4b).

Similarly, the BRM014 treatment also resulted in marked reductions in chromatin accessibility measured by DNase-seq signals (Fig. 4c).

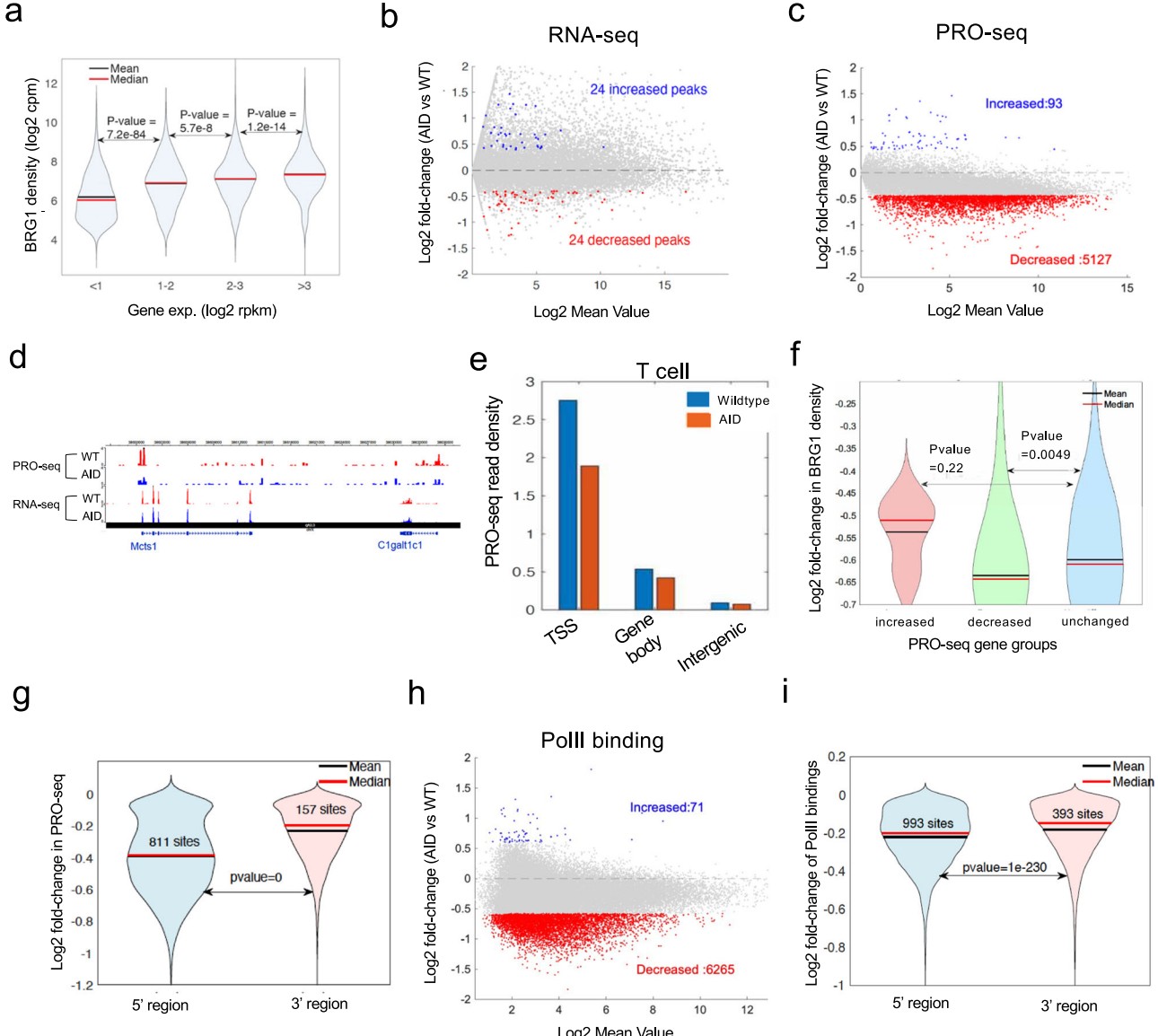

**Fig. 2 | BRG1 positively regulates nascent transcription by facilitating RNA polymerases II binding to chromatin. a** Violin plots show a positive correlation between BRG1 TSS binding and gene expression in wild-type T cells, with 11,206 peaks assessed by a two-sided Wilcoxon rank sum test. **b** MA plot of RNA-seq shows gene expression changes from 17,595 genes after 10-h BRG1 depletion. Red and blue dots mark significant expression decreases and increases; a black line indicates no change. Significance was determined by a two-sided ROTS test with $P < 0.05$ and fold-change >1.5. **c** MA plot of PRO-seq data shows differential nascent transcription in 35,976 genes post 10-h BRG1 depletion. Red (decreased) and blue (increased) dots indicate significant gene changes, with a horizontal black line showing no change, as identified by a two-sided ROTS test ($P < 0.05$, fold-change >1.3). **d** Genome browser snapshot compares PRO-seq and RNA-seq between wild-type (red) and BRG1-depleted (blue) cells. **e** PRO-seq read density compared at TSS, gene body, and intergenic areas between wild type (blue) and AID (red) samples, normalized by region size. **f** Violin plots show fold-change in BRG1 TSS binding across genes with differential transcription, analyzed by a two-sided Wilcoxon rank sum test. Groups are increased (6), decreased (395), and unchanged (249 genes; $P > 0.5$, log2 fold-change < 0.05), based on proximity within ± 2 kb of the TSS. **g** Violin plots illustrate PRO-seq signal fold-changes at 5' (within 1000 bp of TSS) and 3' (within 1000 bp of TTS) regions between BRG1-depleted and wild-type cells, with 881 and 157 sites showing decreased signals after BRG1 depletion, respectively, assessed by a two-sided Wilcoxon rank sum test. **h**, MA plot shows Pol II binding variations across 42,467 sites following BRG1 depletion, with red and blue dots marking sites with significant decreases and increases, respectively, determined by a two-sided ROTS test at $P < 0.05$ and fold-change >1.3. **i** Violin plot shows Pol II binding changes at 5' (within 1000 bp of TSS) and 3' (within 1000 bp of TTS) after 10-h BRG1 depletion, with 993 and 393 fewer sites at 5' and 3', respectively, in BRG1-depleted cells versus wild type, assessed by a two-sided Wilcoxon rank sum test.

Globally, 5794 DHSs showed significantly decreased accessibility, while only 58 DHSs showed increased accessibility upon BRM014 treatment (Fig. 4d). Finally, we compared the changes of chromatin accessibility caused by BRG1-depletion and BRM014 treatment. The analysis revealed a high degree of similarity in chromatin accessibility changes caused by BRG1 depletion and BRM014 treatment (Fig. 4e), providing further evidence for the significant role of BRG1 in maintaining chromatin accessibility.

## Acute depletion of BRG1 results in decreased enhancer activities

Since acute depletion of BRG1 leads to decreased accessibility at a large fraction of DHSs, we next investigated whether it also compromises enhancer activities by measuring p300 binding and H3K27ac levels at these sites. Our ChIC-seq data revealed marked decreases of H3K27ac (Supplementary Fig. 4band Fig. 5a) and p300 (Fig. 5a) signals at a fraction of potential enhancers after acute depletion of BRG1. The analysis revealed 806 (3.8%) significantly decreased and 90 increased

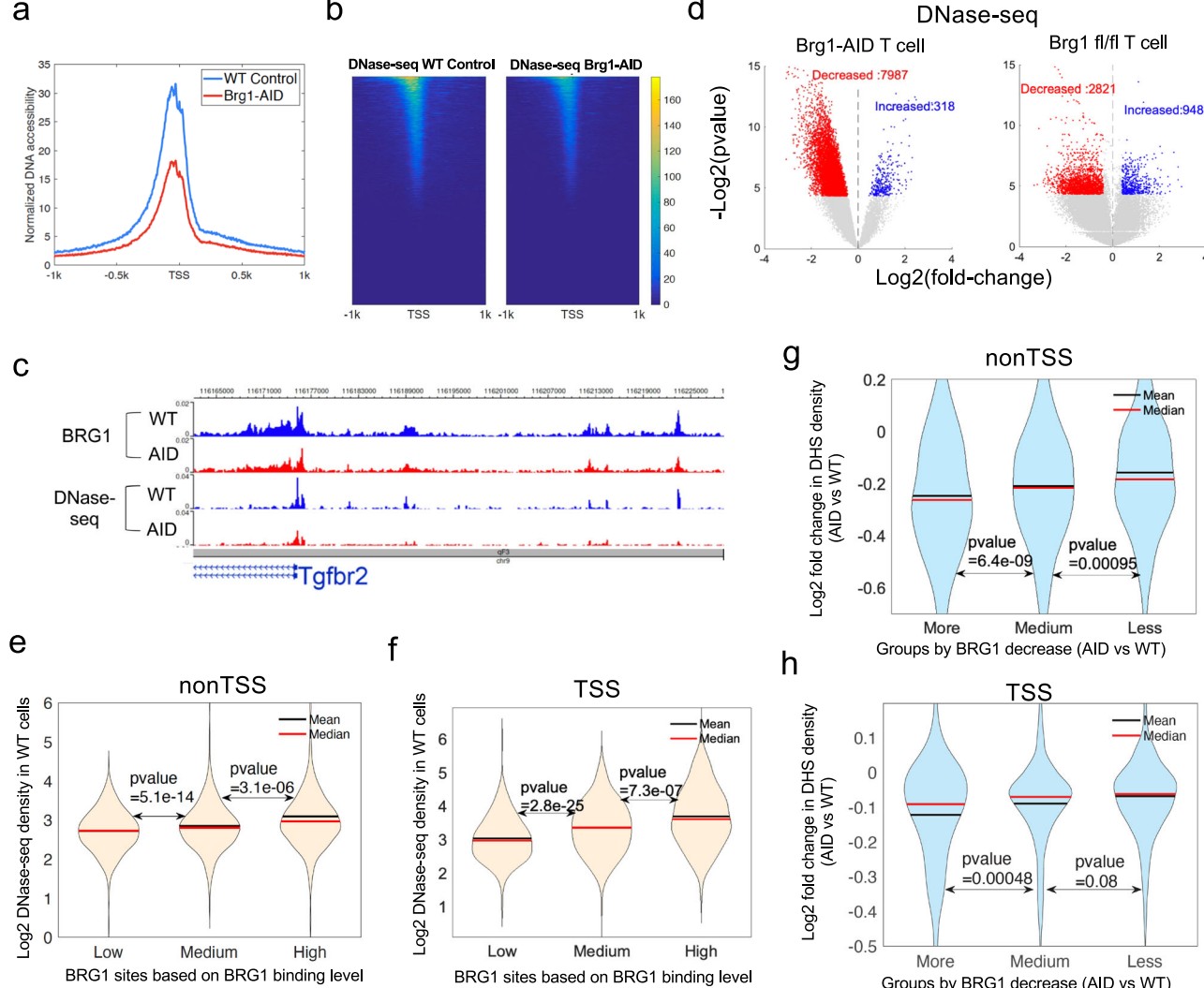

**Fig. 3 | Acute depletion of BRG1 results in significantly decreased chromatin accessibility at BRG1 binding sites in mouse T cells. a** Average profiles of the normalized DNase-seq tag density around TSSs in BRG1-depleted and wild-type cells. Cells depleted of BRG1 for 10 h as described in Fig. 1a were subjected to DNase-seq assay. The red line refers to the BRG1-depleted sample and the blue line refers to wild wild-type sample. The total number of DHSs identified in wild-type and AID CD4 + T cells is 37,572. **b** The DNase-seq signal heatmap for both BRG1-depleted and wild-type samples based on the order ranked by the DNase-seq reads density at the 42,404 TSSs. The color level indicates the DNase-seq reads density level. **c** Genome browser view of ChIC-seq BRG1 binding and DNase-seq chromatin accessibility profiles, with blue for wild type and red for BRG1-depleted cells. **d** Volcano plots depict DNase-seq read density shifts at DHSs following acute BRG1 depletion (left) and conditional Brg1 deletion (right) in T cells. Blue and red dots indicate significant DHS increases and decreases, respectively, assessed by a two-sided ROTS test (*P* < 0.05, fold-change>1.3). **e**, **f**, Violin plots compare chromatin accessibility among low, medium, and high BRG1 binding groups in wild-type T cells at nonTSS (> 5 kb from TSS, panel (**e**)) and TSS (≤ 2 kb from TSS, panel (**f**)) regions, using only DHSs that overlap with BRG1 peaks. Group separation was based on BRG1 binding intensity, with P values determined by two-sided Wilcoxon rank sum tests. **g**, **h**, Violin plots illustrate chromatin accessibility changes at non-TSS (**g**) and TSS (**h**) DHSs in T cells post 10-h acute BRG1 depletion. DHSs were categorized by the extent of BRG1 binding reduction (more, medium, less) compared between BRG1-depleted and wild-type cells, with P values from two-sided Wilcoxon rank sum tests.

H3K27ac peaks (Fig. 5b); meanwhile 5136 (13%) p300 sites showed significantly decreased binding and 258 (0.7%) p300 sites showed increased binding in BRG1-depleted cells (Fig. 5c). To test how H3K27ac modification and p300 binding correlate with BRG1 binding on chromatin, we examined H3K27ac and p300 signal densities at non-TSS and TSS sites which were grouped according to BRG1 binding levels. The results indicated that higher H3K27ac and p300 signal densities were correlated with higher BRG1 binding levels at both non-TSS and TSS sites (Fig. 5d, f). Further, more decreases in H3K27ac and p300 signals were correlated with more decreases in BRG1 binding after auxin-induced BRG1 depletion, particularly at the non-TSS sites (Fig. 5e, g). These results indicated that acute depletion of BRG1 compromises enhancer activities by decreasing chromatin accessibility and p300 binding at enhancers.

## Analyzing chromatin accessibility and gene expression patterns in *Brg1* gene deletion experiments

*Brg1* gene deletion using traditional conditional knockout strategies led to both decreases and increases in DHS accessibility[8,9,12,21,31]. Since traditional gene deletion strategies require a longer time for complete gene deletion, we reasoned that the length of time that gene is deleted may affect the result and decided to test how prolonged absence of BRG1 affects the chromatin accessibility. Because the acute BRG1-depleted cells don't survive more than 24 h, we chose to delete the *Brg1* gene by treating T cells, which were isolated from ERT-Cre/*Brg1*fl/fl mice, with tamoxifen for 3 days (Supplementary Fig. 5a, b). The *Brg1* deletion in T cells resulted in changes of 3769 DHSs (10.0%), including 2821 decreased DHSs (75.0%), which is lower compared to the acute depletion of BRG1 (96.0%), and 948 increased DHSs (25.2%), which is

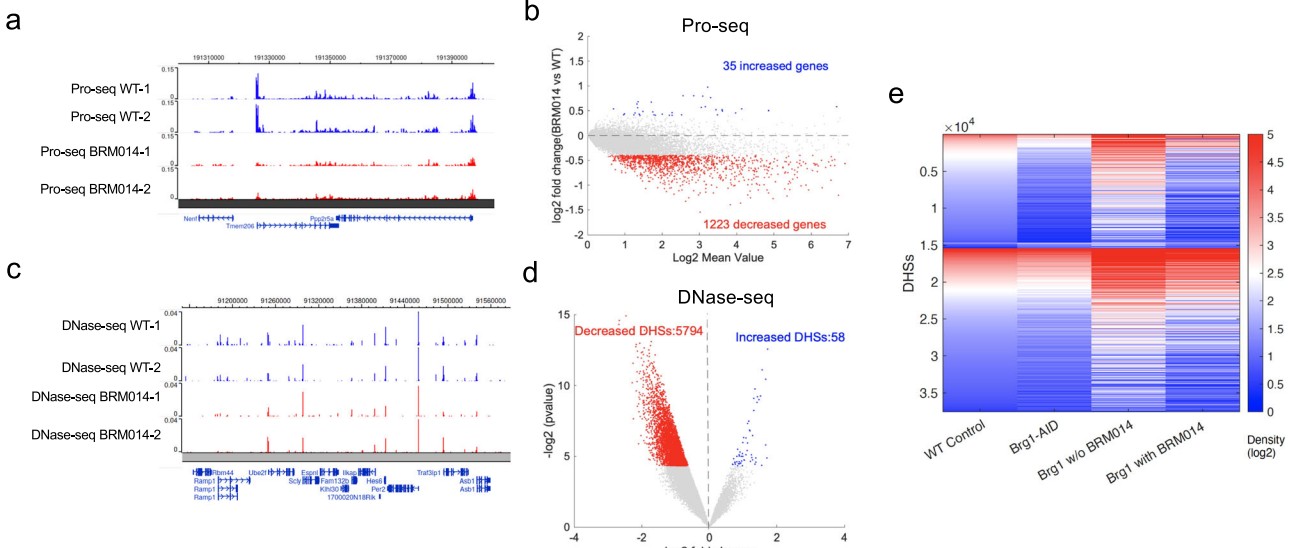

**Fig. 4 | Treated CD4 + T cells with or without BRM014 for 10 h. a.** A snapshot from the genome browser, showing Pro-seq data for both WT (represented by two replicates in blue) and BRM014 (indicated by two replicates in red). **b** MA plot shows PRO-seq derived changes in nascent transcription in T cells after 12-h BRM014 treatment. Blue and red dots represent genes with significant increases and decreases in expression, respectively, with a dashed line for no change. Significance was determined by a two-sided ROTS test using $P < 0.05$ and fold-change > 1.3 criteria for differential expression. **c** A snapshot from the genome browser, showing DNase-seq data for both WT (represented by two replicates in blue) and BRM014 (indicated by two replicates in red). **d** A volcano plot displays DNase-seq density changes at DHSs after 12-h BRM014 treatment. Blue and red dots show significant increases and decreases, respectively; the dashed line marks no change. DHSs were selected based on $P < 0.05$ and fold-change > 1.3, using a two-sided ROTS analysis. **e** A heatmap shows DNase-seq read density at DHSs for both BRG1-depleted samples and samples with a 12-h treatment with the BRM014 inhibitor. The DHSs are separated into three groups (decreased, increased, and unchanged) based on the changes in DNase-seq read density upon BRG1-depletion (AID versus WT).

substantially more than that caused by the acute BRG1 depletion (4.0%) (Supplementary Fig. 5ca and Fig. 3d). Thus, it appears that prolonged BRG1 absence leads to increased accessibility at a higher fraction of changed regulatory regions, which could compensate for the loss of BRG1 and increase cell survival. Indeed, analysis of the RNA-seq data from wild-type and *Brg1* gene-deleted CD4 + T cells revealed that the up-regulated genes in *Brg1* knock-out T cells were significantly enriched in the gene ontology term of negative regulation of cell death (Supplementary Fig. 5d), which included cell cycle regulators, signaling molecules, and transcription factors and could be involved in the cell survival in the absence of BRG1. Although the chromatin accessibility at TSSs or enhancers of these genes was compromised by the acute depletion of BRG1, it remained unchanged or even became higher in the Brg1^{fl/fl} gene knockout cells as exemplified by *Rarg* and *Ccr7* genes (Supplementary Fig. 5e).

To explore other potential mechanisms that could potentially compensate for the loss of BRG1 in the cells, we examined the expression of *Smarca2*, an important paralog of the *Brg1* gene, and 24 well-established and potential subunits of the BAF complexes in wild type and *Brg1*-deleted CD4 + T cells. The gene expression of *Brg1* displayed a consistent reduction at both the 60-h and 6-day intervals (Supplementary Fig. 6). Notably, only Smarca2 consistently showed increased expression levels across all replicates and time points in the *Brg1*-deleted cells (Supplementary Figs. 6–8). This observation suggests that increased BRM expression could potentially serve as a compensatory mechanism in response to the prolonged loss of Brg1 in the cells.

### Acute depletion of BRG1 results in genome-wide decreases in chromatin accessibility and nascent transcription in hepatocytes and fibroblasts

To further validate the results observed in T cells, we also acutely depleted BRG1 in hepatocytes and fibroblasts isolated from the BRG1-AID knock-in mice (Supplementary Fig. 9a). Examination of the BRG1 peak distribution revealed substantial variations across different cell

types. Specifically, while TSS peaks accounted for 46% of BRG1 binding sites in T cells (Fig. 1d), this percentage was notably reduced to 34% in hepatocytes (Supplementary Fig. 9b) and further decreased to 23% in fibroblasts (as depicted in Supplementary Fig. 9b). This observed disparity among cell types underscores the dynamic nature of BRG1's binding profile. Importantly, this variation highlights the suitability of hepatocytes and fibroblasts as valuable cell models for investigating BRG1's functional roles, providing insight into how its functions may vary across diverse cellular contexts.

The acute BRG1 depletion in hepatocytes led to significantly changed accessibility of 14,461 DHSs (29.0%). Consistent with the results in T cells, the vast majority of changed DHSs (14,371 sites, 99.4%) showed decreased accessibility while only 90 DHSs (0.6%) showed increased accessibility by the acute BRG1 depletion (Fig. 6a and Supplementary Fig. 9c). Further, conditional deletion of the *Brg1* gene in hepatocytes led to a total of 5568 changes DHSs, including 4301 decreased (77.2%) and 1267 increased (22.8%) DHSs (Fig. 6b and Supplementary Fig. 9d, e). The auxin treatment-induced BRG1 depletion in fibroblast cells led to significantly decreased BRG1 binding at 14,371 (17.7%) BRG1 peaks (Fig. 6c), which overlapped with 59% TSSs and 26% potential enhancers. Similar to the results from T cells and hepatocytes, the acute BRG1 depletion also resulted in mainly decreases in chromatin accessibility in fibroblasts (Fig. 6d).

To examine if the nascent transcription in hepatocytes and fibroblasts is also compromised by acute depletion of BRG1, we performed PRO-Seq in these cells. Similar to CD4 + T cells, the TSS regions exhibited elevated PRO-Seq signals in hepatocytes and fibroblasts and showed decreases of nascent transcripts after the BRG1 depletion (Supplementary Fig. 9f). Both hepatocytes and fibroblasts showed global trends of decrease in PRO-seq signals by acutely depleting BRG1. we identified only 1324 significantly decreased genes in hepatocytes, while there were 42 increased genes (Fig. 6e). This relatively smaller number of significantly changed genes may be related to the quiescent nature of hepatocytes, which is also consistent with the normal

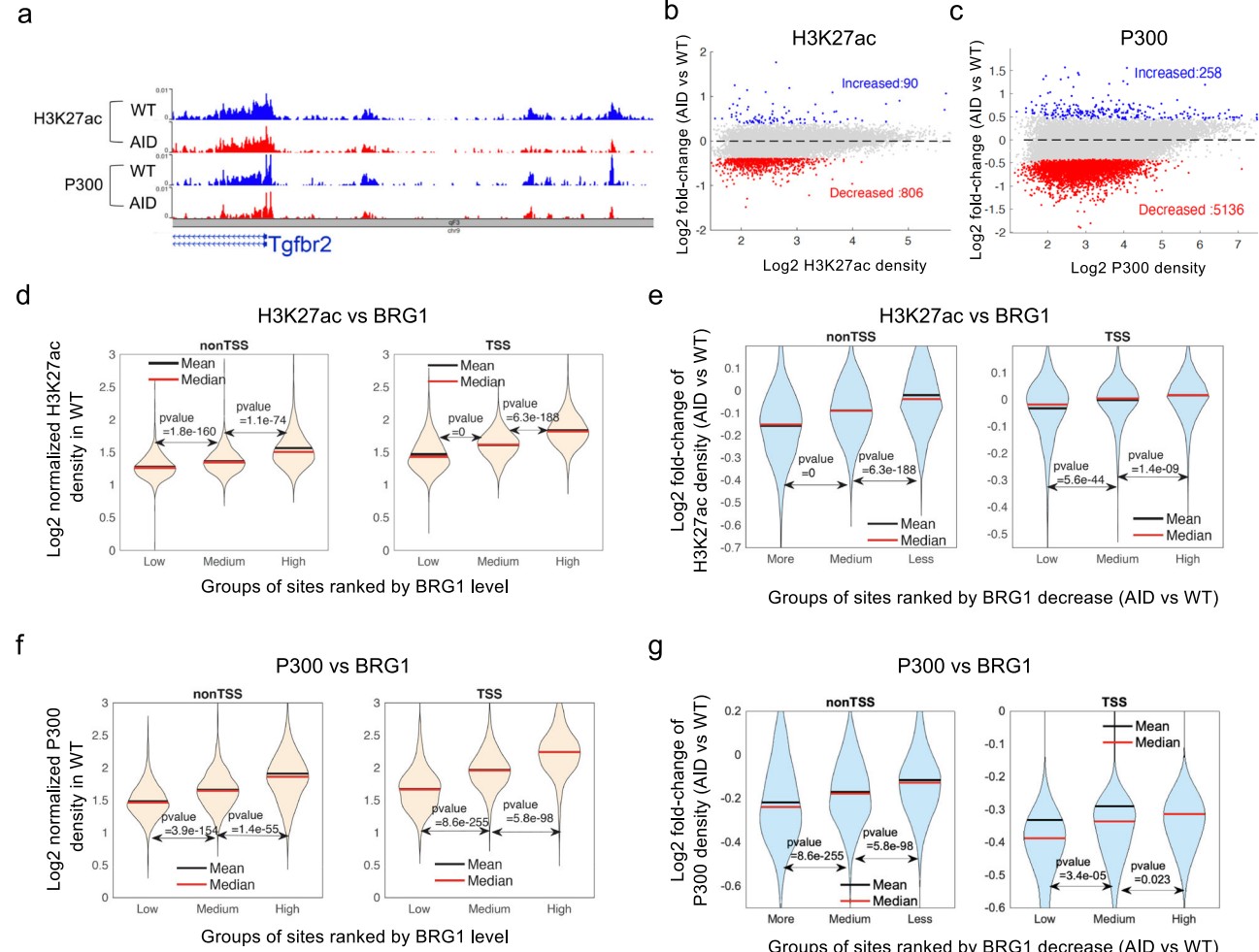

**Fig. 5 | Acute depletion of BRG1 results in significantly decreased p300 binding and H3K27 acetylation at BRG1 binding sites in mouse T cells. a** Genome browser snapshot displays H3K27ac and P300 ChIC-seq profiles, with blue tracks for wild type and red for BRG1-depleted cells. **b** MA plot depicts H3K27ac changes at 21,347 H3K27ac peaks defined in WT cells following 10-h BRG1 depletion in T cells, with blue and red dots marking significantly increased and decreased peaks, respectively, identified using a two-sided ROTS test (*P* < 0.05, fold-change > 1.3). **c** MA plot reveals changes in P300 binding at 39,315 P300 peaks defined in WT cells due to 10-h BRG1 depletion in T cells, with blue and red dots indicating significant increases and decreases, respectively, determined by a two-sided ROTS test (*P* < 0.05, fold-change > 1.3). **d** Violin plots compare H3K27ac levels across BRG1 binding groups (low, medium, high) in wild-type T cells at nonTSS (> 5 kb from TSS) and TSS (≤ 2 kb) regions. Only peaks overlapping with BRG1 were assessed,

categorized by BRG1 levels, with P values determined using a two-sided Wilcoxon rank sum test. **e** Violin plots illustrate H3K27ac alterations at nonTSS (left) and TSS (right) peaks in T cells following 10-h BRG1 depletion. Peaks were grouped by BRG1 binding changes (more decrease, medium, less decrease) compared between BRG1-depleted and wild-type cells, with P values calculated by a two-sided Wilcoxon rank sum test. **f** Violin plots depict variations in p300 binding at nonTSS (> 5 kb from TSS) and TSS (≤ 2 kb) regions among low, medium, and high BRG1 binding groups in wild-type T cells. The analysis included only p300 peaks overlapping BRG1, categorized by BRG1 binding intensity, with P values determined by a two-sided Wilcoxon rank sum test. **g** Violin plots compare p300 alterations at non-TSS (left) and TSS (right) peaks in T cells following 10-h BRG1 depletion, with peaks grouped by the extent of binding decrease (more, medium, less) in BRG1-depleted versus wild-type cells. *P* values were calculated using a two-sided Wilcoxon rank sum test.

morphology of liver in *Brg1* knock out mice using Alb-Cre. Consistent with this hypothesis, the acute BRG1 depletion in the more proliferative fibroblasts resulted in 2611 significantly decreased genes while only 3 increased genes were detected (Fig. 6f). All together, these results indicate that BRG1 contributes critically to the chromatin accessibility, p300 binding, and nascent transcription in multiple cell types.

## Interplays between BRG1 and histone acetylation

To test how BRG1 interplays with histone acetylation in a dynamically regulated system, we examined BRG1 binding profiles and H3K27ac modification after inhibition of histone deacetylases in fibroblasts using TSA treatment. The data revealed that BRG1 binding was substantially increased genome-wide by the TSA treatment for 2 h and the BRG1 peaks showed a substantial expansion to their neighboring regions (Fig. 7a–c). As expected, the TSA-induced increase of BRG1

binding and peak expansion were compromised upon depletion of BRG1 by auxin treatment. As expected, TSA treatment induced a substantial increase in the global levels of H3K27ac. Interestingly, the depletion of BRG1 abolished the TSA-induced H3K27ac in cells. These results indicated that chromatin remodeling by BRG1 interplays with histone modification and BRG1 plays an important role in TSA-induced H3K27ac in the cells.

## Acute depletion of BRG1 results in narrower nucleosome spacing at gene promoters and enhancers

Since BRG1 has nucleosome remodeling activity, alterations in nucleosome organization at regulatory regions may likely contribute to the acute BRG1 depletion-induced decreases in nascent RNAs, DHS accessibility, and enhancer activities. To test this, we analyzed genome-wide nucleosome occupancy by MNase-Seq. Our data confirmed the previous observation that active genes exhibited much

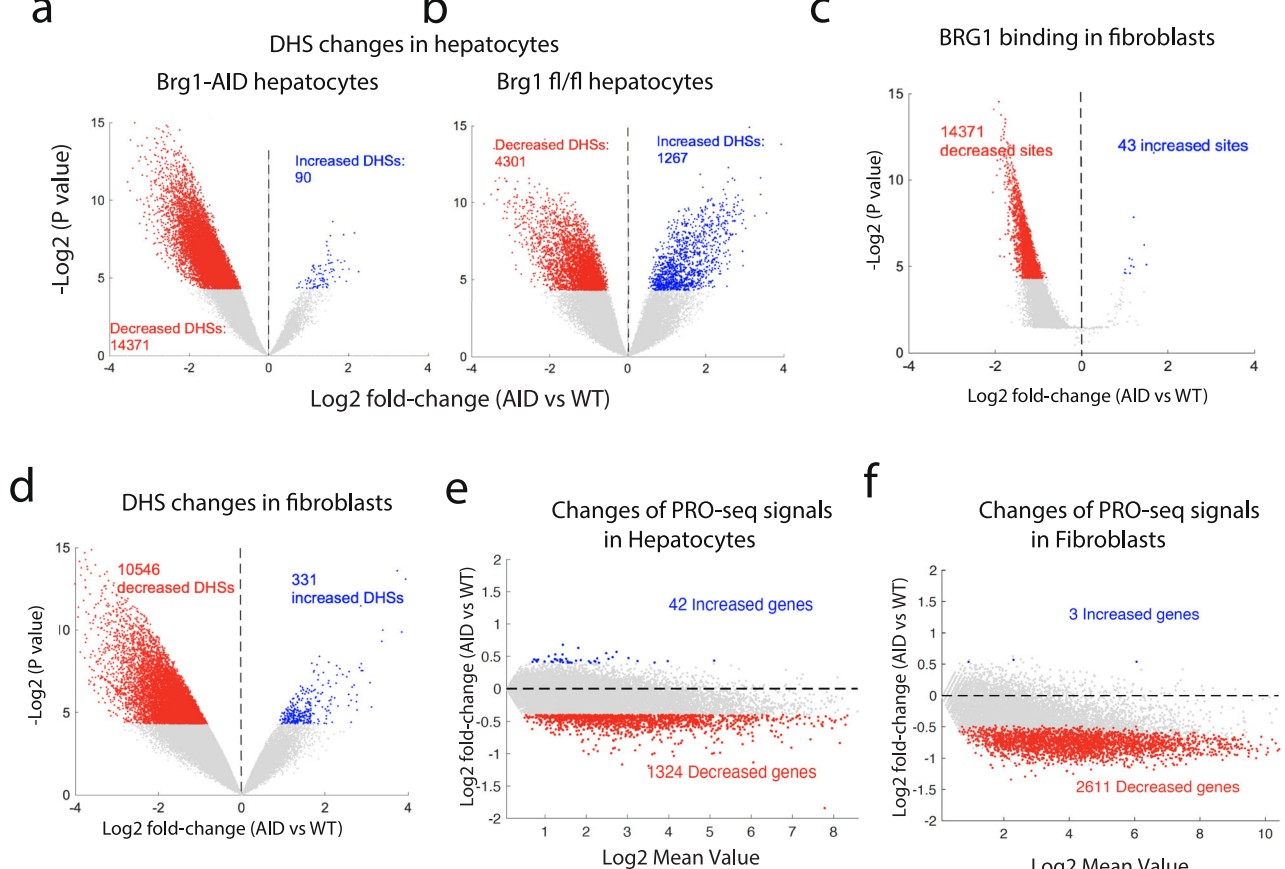

**Fig. 6 | Acute depletion of BRG1 results in decreased chromatin accessibility and nascent transcription in hepatocytes and fibroblasts. a, b** Volcano plots display DNase-seq density changes at 49,891 DHSs in hepatocytes following acute BRG1 depletion (**a**) or conditional Brg1 gene deletion (**b**), with blue and red dots indicating significant increases and decreases, respectively, identified using a two-sided ROTS test (P < 0.05, fold-change> 1.3). **c** Volcano plot depicts changes in 81,017 BRG1 ChIC-seq binding sites in fibroblasts after 10-h auxin-induced depletion. Blue and red dots mark significantly increased and decreased BRG1 peaks, respectively, assessed by a two-sided ROTS test. **d** Volcano plot reveals DNase-seq read density changes at 80,732 DHSs in fibroblasts after acute BRG1 depletion, with blue and red dots showing significant increases and decreases, respectively, determined by a two-sided ROTS test (P < 0.05, fold-change> 1.3). **e, f,** MA plots display PRO-seq-derived changes in hepatocytes (**e**) and fibroblasts (f) after 10-h BRG1 depletion. Blue and red dots mark genes with significant increases and decreases, respectively, with dotted lines for no change. Significance was set at P < 0.05 and fold-change > 1.3, analyzed by a two-sided ROTS test.

better nucleosome phasing surrounding TSSs compared to inactive genes[32] (Supplementary Fig. 10a, b). We did not observe global changes in nucleosome organization after acute BRG1 depletion (Supplementary Fig. 10b, c). Since the −1 and +1 nucleosome positions around TSSs are related to gene transcriptional activity, we focused our analysis on these two nucleosomes in BRG1-depleted and control CD4 + T cells. We identified 918 promoters and 990 enhancers in CD4 + T cells, which exhibited the nucleosome positioning shifts significantly correlated with the decrease of BRG1 binding in these regions (Fig. 8a). We classified these patterns into three groups: at the 189 "Shift 1" TSSs, the +1 nucleosome was shifted upstream by about 36 base pairs on average; at the 180 "Shift 2" TSSs, the −1 nucleosome shifted downstream by about 49 base pairs on average; and at the 549 "Shift 3" TSSs, the −1 nucleosome shifted downstream by 40 base pairs while the +1 nucleosome shifted upstream by 27 base pairs (Fig. 8b, cand Table 1). Similarly, we identified three nucleosome shift patterns at enhancers: at the 176 "Shift 1" enhancers, the +1 nucleosome was shifted upstream by about 36 base pairs on average; at the 386 "Shift 2" enhancers, the −1 nucleosome shifted downstream by about 35 base pairs on average; and at the 428 "Shift 3" enhancers, the −1 nucleosome shifted downstream by 36 base pairs while the +1 nucleosome shifted upstream by 33 base pairs (Fig. 8c and Table 1). All three nucleosome shift patterns resulted in narrower nucleosome spacing between the −1 and +1 nucleosomes around TSSs and enhancers (Fig. 8c). Similarly, we also

identified three nucleosome shift patterns at promoters and enhancers in hepatocytes after the acute depletion of BRG1 (Fig. 8a, lower panel). All three nucleosome shift patterns were significantly correlated with decreased BRG1 binding and decreased accessibility at both the promoter and enhancer regions in T cells and hepatocytes (Fig. 8d, e), which could lead to decreased RNA Pol II binding and compromised transcription. Indeed, we found that all three nucleosome shifts are significantly correlated with the decreased PRO-Seq signals (Fig. 8d, e). Notably, while the Shift 1 and Shift 2 patterns may be associated with significant decreases in DNase-seq, H3K27ac, p300 and PRO-Seq signals at either TSSs or enhancers, Shift 3 patterns are significantly correlated with decreases in these signals at both TSSs and enhancers (Fig. 8d, e), suggesting more nucleosome shifting toward the enhancer center caused by the BRG1 depletion may be associated with more severe consequence in compromising chromatin modification, p300 binding and transcription.

We observed that the DNA contents for the TSS regions with shifted nucleosomes in T cells contained significantly lower CpG and GC contents compared to the TSSs with non-shifted nucleosomes (Supplementary Fig. 10d), which is consistent with a previous observation that the SWI/SNF-dependent genes are often non-CpG-island genes[33]. Although the same trends were observed in hepatocytes, the differences in hepatocytes were only modest and not statistically significant (Supplementary Fig. 10e).

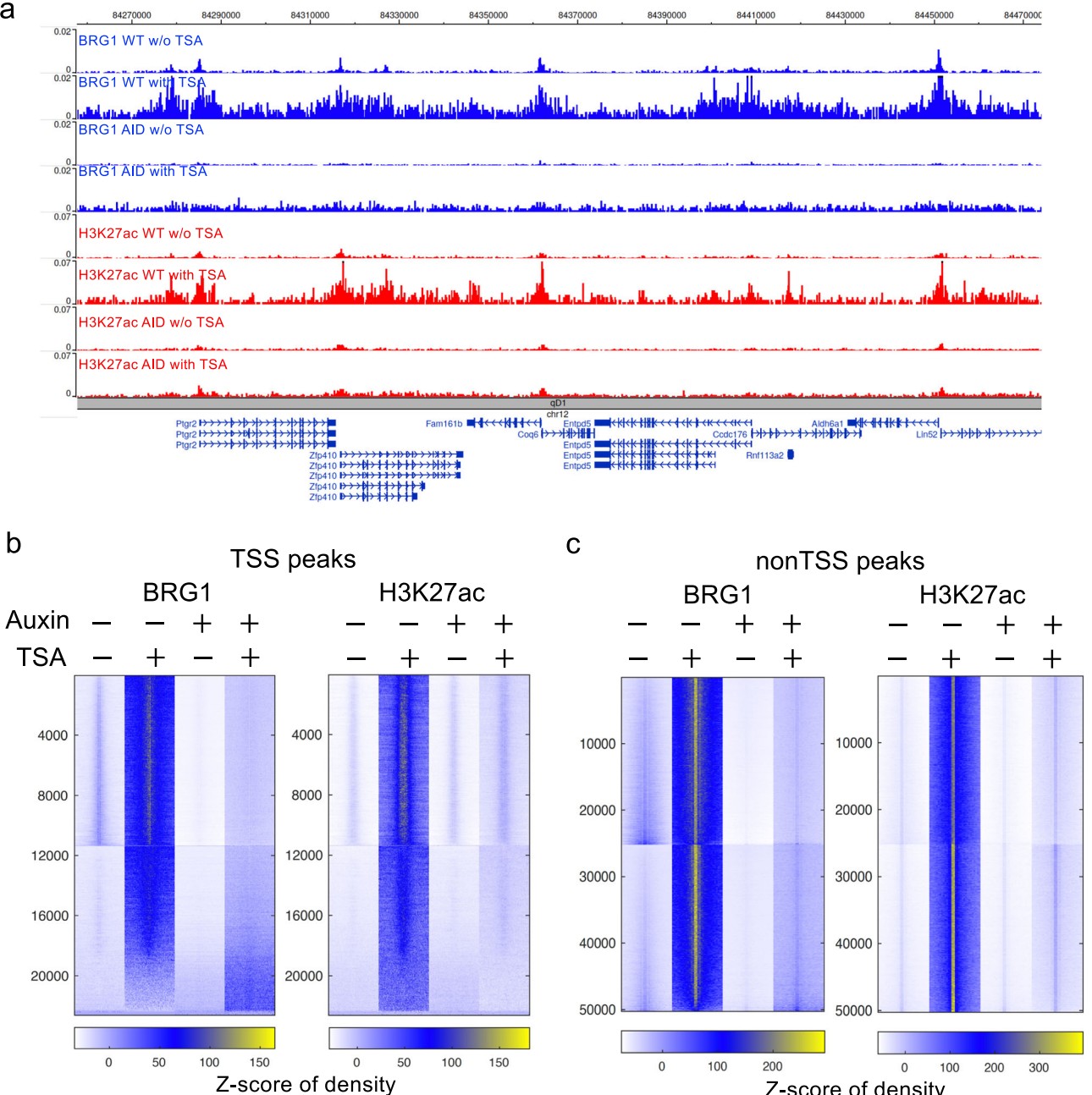

**Fig. 7 | BRG1 interplays with post-translational histone modification. a** Genome Browser snapshots showing the BRG1 binding profiles (top four blue tracks) and H3K27ac profiles (bottom four red tracks) in WT control cells without TSA treatment, WT with TSA treatment, AID cells without TSA treatment, and AID cells with TSA treatment. The WT or AID cells were treated with auxin for 10 h, followed by TSA treatment for 2 h. **b** Heatmaps of BRG1 density (**left panel**) and H3K27ac density (**right panel**) around TSS regions in WT control cells with or without TSA treatment as well as AID BRG1-depleted cells with or without TSA treatment. **c** Heatmaps of BRG1 density (**left panel**) and H3K27ac density (**right panel**) around non-TSS regions in WT control cells with or without TSA treatment as well as AID BRG1-depleted cells with or without TSA treatment.

Furthermore, we performed functional annotation for the genes associated with the three shift patterns using DAVID and GREAT, separately[34,35]. The analysis revealed that the majority of enriched categories related to the three shift patterns at enhancers were relevant to T cell development and function, especially for Shift 2 and Shift 3 (Supplementary Fig. 11). However, many of the significantly enriched terms of the three TSS shift patterns are related to house-keeping activities such as chromosome organization, regulation of growth rate, metabolic process etc. (Supplementary Fig. 12). Together, these results show that the acute depletion of BRG1 alters the nucleosome occupancy patterns at TSSs and enhancers; while the former is enriched in

house-keeping genes, the latter is enriched in T cell-specific functions. Those results are consistent with the notion that enhancers are more related to cell type-specific functions.

## Acute depletion of BRG1 compromised nucleosome phasing surrounding key TF motifs

Since enhancers are often bound by sequence-specific transcription factors, which play critical roles in cell differentiation and function. To investigate how the depletion of BRG1 and mis-regulation of nucleosome structure may be related to sequence-specific transcription factors, we examined the TF motifs enriched in the enhancer regions

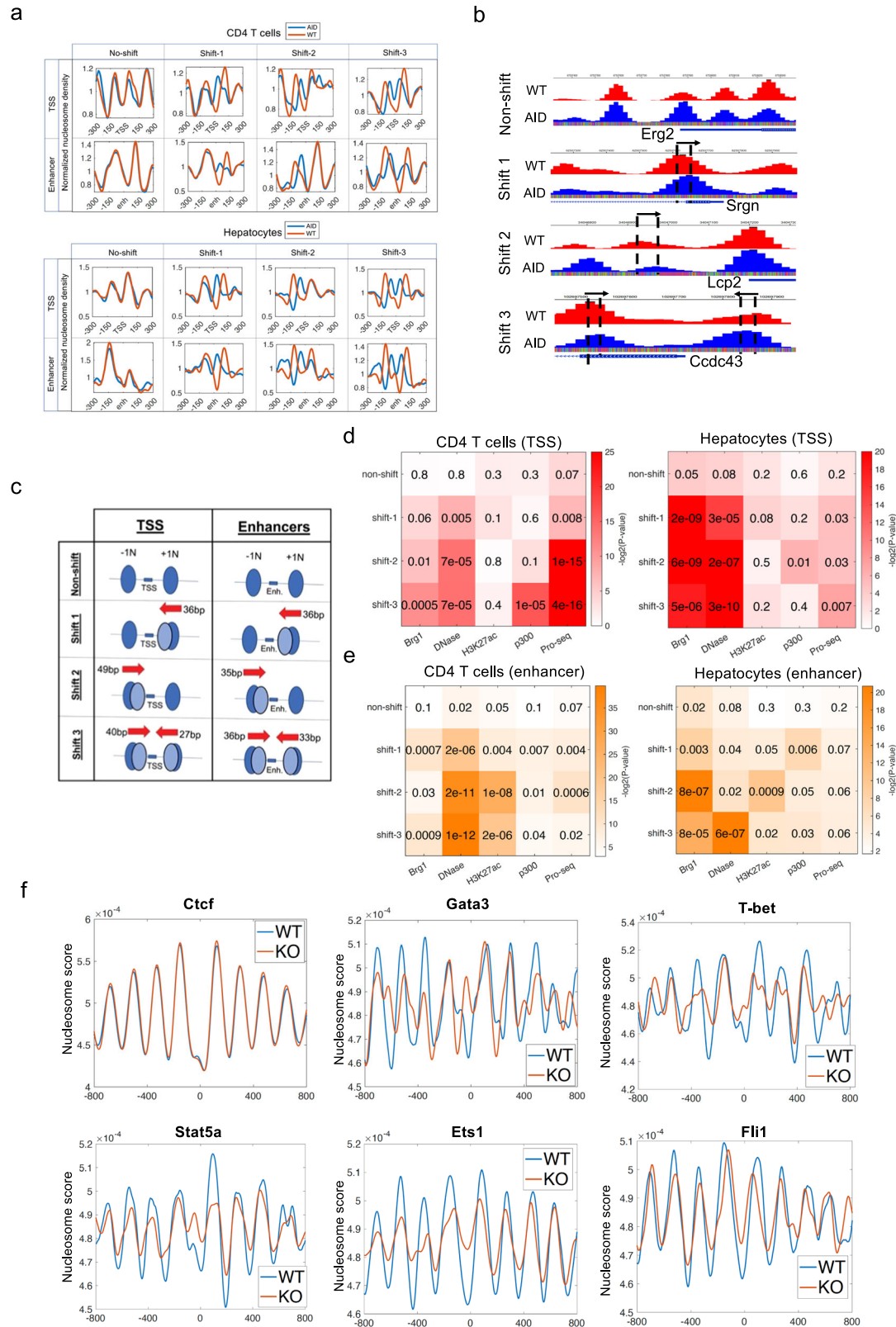

with shifted nucleosomes. The analysis revealed that TF motifs such as AP1, Nfatc1, and ETS, which are implicated in T cell differentiation and function in the TSS and enhancer regions associated with the three nucleosome shift patterns after the acute BRG1 depletion in T cells (Supplementary Fig. 13).

To further test how BRG1 depletion alters the nucleosome organization at TF motif regions, we specifically examined the nucleosome

organization patterns surrounding motifs of several key TFs in CD4 + T cells. The ubiquitously expressed chromatin binding protein CTCF is an important regulator for three-dimensional chromatin structure and is associated with well-phased nucleosomes surrounding its binding motifs[32,36]. We found that the nucleosome phasing and occupancy patterns surrounding the CTCF motif are highly similar between control and BRG1-depleted CD4 + T cells, suggesting that

**Fig. 8 | Acute depletion of BRG1 results in narrower nucleosome spacing at TSSs and enhancers. a** Acute BRG1 depletion in T cells (upper panel) and hepatocytes (lower panel) led to four nucleosome shifting patterns at TSSs or enhances, analyzed by MNase-seq. Profiles at TSSs or enhancers show 'non-shift' (no movement), 'Shift 1' (+1 nucleosome upstream), 'Shift 2' (−1 nucleosome downstream), and 'Shift 3' (+1 upstream and −1 downstream shift). Counts for each pattern are as follows: T cell TSSs (196 non-shift, 189 shift-1, 180 shift-2, 549 shift-3), T cell enhancers (287, 176, 386, 428), hepatocyte TSSs (275, 510, 484, 904), and hepatocyte enhancers (91, 201, 181, 395). **b** Genome browser snapshot displays four nucleosome shifting patterns around T cell TSSs: blue tracks for nucleosome positions in BRG1-depleted cells, red for wild type. **c** Cartoon schemes for nucleosome shifting patterns of "shift 1, shift 2, and shift 3" at TSSs and enhancers.

**d** TSSs with "non-shift", 'Shift 1', 'Shift 2', and 'Shift 3' patterns correlate with significant reductions in BRG1, DNase-seq, H3K27ac, p300, and PRO-seq signals post-BRG1 depletion in CD4 + T cells and hepatocytes. Changes at these TSSs were quantified by comparing datasets between BRG1-depleted and wild-type cells, with color intensity reflecting the negative log P values from a one-sided hypergeometric test. **e** Enhancers with 'Shift 1', 'Shift 2', and 'Shift 3' patterns in CD4 + T cells and hepatocytes show marked reductions in BRG1 binding, DNase-seq, and PRO-seq signals after BRG1 depletion. The intensity of colors corresponds to the negative log P values, determined by a one-sided hypergeometric test. **f** Nucleosome profiles surrounding key transcription factor binding motifs in wild type and BRG1-depleted CD4 + T cells. The blue and red lines represent wild-type control and BRG1 depletion cells, respectively.

## Table 1 | MNase-seq detects different nucleosome shifting patterns surrounding enhancers and TSSs

|  | Location of −1 nucleosome (WT) | Location of +1 nucleosome (WT) | Spacing (WT) | Location of −1 nucleosome (AID) | Location of +1 nucleosome (AID) | Spacing (AID) | Shifting at −1 Nucleosome | Shifting at +1 Nucleosome |
|---|---|---|---|---|---|---|---|---|
| Enhancers |  |  |  |  |  |  |  |  |
| Shift-1 | −88 | 92 | 180 | −88 | 56 | 144 | 0 | −36 |
| Shift-2 | −102 | 71 | 173 | −67 | 71 | 138 | 35 | 0 |
| Shift-3 | −102 | 92 | 194 | −66 | 59 | 125 | 36 | −33 |
| TSSs |  |  |  |  |  |  |  |  |
| Shift-1 | −93 | 70 | 163 | −93 | 34 | 127 | 0 | −36 |
| Shift-2 | −108 | 56 | 164 | −59 | 56 | 115 | 49 | 0 |
| Shift-3 | −105 | 70 | 175 | −65 | 43 | 108 | 40 | −27 |

BRG1 is dispensable for the binding of CTCF to chromatin (Fig. 8f). However, the nucleosome organization patterns surrounding motifs of several key T cell-specific TFs, including Gata3, T-bet, Stat5a, Ets1, and Fli1, are substantially altered, by either compromised nucleosome depletion over the motifs or shifting of flanking nucleosomes toward the motifs (Fig. 8f). These results may explain the extensive compromises of chromatin accessibility and nascent transcription upon depletion of BRG1.

## Discussion

In this study, we tagged BRG1, the essential ATPase subunit of BAF complexes, with AID in mice and demonstrated rapid depletion of the AID-tagged BRG1 in isolated CD4 + T cells, fibroblasts and hepatocytes. Analysis of nascent RNAs using PRO-Seq revealed changed expression of 5220 genes by the acute depletion of BRG1 in CD4 + T cells. Among the changed genes, the vast majority (98.2%) showed decreased expression and only a minor fraction showed increased expression (1.8%) by the BRG1 depletion. Similar trends of changes in nascent transcription were observed in fibroblasts and hepatocytes after acute depletion of BRG1. The decreased genes included both housekeeping genes and cell-specific genes, consistent with the binding profiles of BRG1 on chromatin. Further, the decrease in gene expression positively correlated with the loss of BRG1 binding on chromatin upon BRG1 depletion. Thus, these data support a model that the primary function of BRG1 is transcriptional activation. However, this model is apparently in contrast with the previous observations that BRG1 both represses and activates genes using the gene knockdown or deletion assays[13,15,21–26]. Several possibilities could account for the discrepancy: (1) indirect and secondary target genes of BRG1 may increase during the long-term deletion of *Brg1* by the traditional gene deletion strategy; (2) it may take long time to activate the BRG1-repressed genes in differentiated cells, such as those of Polycomb target genes; (3) during the long-term deletion of *Brg1* by the traditional gene deletion strategy, the cells may develop mechanisms that could compensate for the absence of BRG1, especially for house-keeping genes required for cell survival. Indeed, the pathway that negatively regulates cell death was identified in the up-regulated genes detected by RNA-seq analysis of

CD4 + T cells with conditional *Brg1* gene deletion. BRG1 was found to colocalize with other ATP-dependent chromatin remodelers[8] and thus its activity could be partially compensated for by other remodelers for long-term cell survival. Indeed, long-term *Brg1* deletion is associated with increased transcription of *Smarca2*, an important paralog of the *Brg1* gene, which could be the potential compensatory mechanism as suggested[13,37,38].

How does BRG1 facilitate nascent transcription? Our data suggest that BRG1 mainly facilitates nascent transcription by promoting RNA Pol II binding to chromatin because decreased Pol II binding is positively correlated with the loss of BRG1 binding upon acute BRG1 depletion. This function of BRG1 is most likely mediated by its activity to maintain the accessibility of gene enhancers and promoters, which critically contributes to the binding of p300, H3K27 acetylation, and nascent transcription. Further, our observation that TSA treatment increased BRG1 binding and depletion of BRG1 severely compromised TSA-induced histone acetylation suggests a strong interplay between ATP-dependent chromatin remodeling activity and posttranslational histone modifications. Our data not only reinforced the notion that BRG1 plays an important role in establishing enhancer landscapes in cells[9,15,26], but also uncovered that BRG1 positively contributes to gene nascent transcription via its binding. Overall, our results are consistent with a hypothesis that BRG1 regulates either chromatin accessibility or chromatin factor binding or both, which are important for the function of underlying enhancers. Several recent papers also presented evidence that the continuous presence of SWI/SNF complexes is required for chromatin accessibility at transcription regulatory regions on chromatin[39–41]. However, different from our observations in the differentiated cell types, Weber et al. found that mSWI/SNF promotes transcriptional repression by redistributing the Polycomb repressive complexes in murine embryonic stem cells[42]. Further clarification is needed on whether mSWI/SNF is involved in the repression by the Polycomb repressive complexes in fully differentiated cells.

As the essential ATPase subunit of the BAF complexes, BRG1 plays crucial roles in creating and/or maintaining accessible chromatin at regulatory regions by nucleosome sliding and/or eviction[5,7,14,43]. Thus, the changed chromatin structure and nucleosome positions could

reveal the underlying molecular mechanisms of BRG1's function. In this study, we uncovered three different nucleosome shift patterns both at promoter and enhancer regions (Shift 1, Shift 2, and Shift 3), which might be generated by the distinctly enriched sequence-specific transcription factors. All three shift patterns resulted in narrower spacing between the neighboring nucleosomes both at TSSs and enhancers, consistent with decreased accessibility and nascent transcription from these elements. Finally, our data provide direct mechanistic support for working models for RSC complex and SWI/SNF complex functions proposed in recent structural studies[20,44].

In summary, by acute depletion of BRG1, our study suggests that BRG1 primarily functions genome-wide as an activator of gene expression by maintaining chromatin accessibility at gene TSSs and enhancers. Our results with BRG1 also imply that functions of many other genes based on long-term gene deletion studies may need to be reevaluated and the acute protein depletion using Degron strategies is powerful in revealing immediate and direct gene functions.

## Methods

### Brg1-AID mice, Brg1 fl/fl x Alb-Cre mice, Brg1fl/fl x CAG-Cre/ERT2 mice

The mice were purchased from the Jackson Laboratory. The Brg1-AID mice were generated using CRISPR/Cas9-mediated knockin[45]. Briefly, a guide RNA (sgRNA: GGTCAGGACTCAGGAATGTC) was designed to cut 7 bp downstream of the Brg1 stop codon and was synthesized using Thermo Fisher's sgRNA custom in vitro transcription services. A single-strand oligonucleotides purchased from Integrated DNA Technologies (TCAGGACCGCTCAGGAAGTGGCAGTGAGGAAGAC**CCT AAAGATCCA GCCAAACCTCCGGCCAAGGCACAAGTTGTGGGATGGCCACCGGTG AGATCATACCGGAAGAACGTGATGGTTTCCTGCCAAAAATCAAGC GGTGGCCCGGAGGCGGCGGCGTTCGTGAAG**TGAAGAAGACATTCC TGAGTCCTGACCCCGAGGC; the sequence in bold encodes the AID tag while the flanking sequences are the homologous arms) was used as a donor for mediating homology-directed repair (HDR). To generate AID knockin mice, the sgRNA (20 ng/µl), Cas9 mRNA (50 ng/µl), and donor oligonucleotides (100 ng/µl) were co-microinjected into the cytoplasm of fertilized mouse eggs. The injected embryos were cultured overnight in M16 medium (Sigma) overnight in a 37 °C incubator with 6% $CO_2$. Embryos that reached the 2-cell stage of development were implanted into the oviducts of pseudopregnant foster mothers (CD-1 mice from Charles River Laboratory). Offspring born to the foster mothers were genotyped by PCR and Sanger sequencing and founder mice with desired knockin were bred with C57BL/6 mice to expand the colonies. Brg1 fl/fl x Alb-Cre mice, and Brg1fl/fl x CAG-Cre/ERT2 mice were bred and maintained in NIEHS animal facility[3,46]. This study was approved by the National Institute of Health and complied with all relevant ethical regulations with the mouse protocol numbers H-0141R3, and H-0141R5. Three to five 6–10 weeks old mice were used for each experiment. The mice sex and gender were not considered in this study because the gender didn't affect the results.

The following primers were used for genotyping the mice:

For Brg1-AID mice: the forward primer 5′-TCACAAACTAT TCCCCCTGTTC and the reverse primer 5′-GCCTAAAAATGTGGAA GATTGC were used for PCR amplification. The PCR products for wild type and Brg1-AID knock-in alleles were 197 bp and 329 bp, respectively (Supplementary Fig. 1b).

For Brg1 fl/fl mice, three primers P1, P2, P3 (primer P1, GTCAT ACTTATGTCATAGCC; primer P2, GCCTTGTCTCAAACTGATAAG; primer P3, GATCAGCTCATGCCCTAAGG) were used as previously described[3].

For genotyping the Cre transgene: the forward primer 5′-GC GGTCTGGCAGTAAAAACTATC, and reverse primer 5′- GTGAAACAG CATTGCTGTCACTT were used. The primers used in this study were ordered in Integrated DNA Technologies (IDT).

## Isolation and culture of T cells, hepatocytes and fibroblasts

Naive CD4 + T cells were purified from lymph nodes of Brg1-AID mice, Brg1^fl/fl x CAG-Cre/ERT2 mice, and control mice by magnetic beads using EasySep Mouse CD4 + CD62L + T cell Isolation Kit (Stem Cell) according to the manufacturer's instruction were sorted by flow cytometry on a FACSAria II cell sorter (BD Biosciences) for CD4 + CD8−CD62LhighCD44low cells. The purity of naive CD4 + T cells isolated by beads was assessed by flow cytometry for CD4 +, CD8 −, CD44-low and CD62L-high and over 98% of purity were considered for further experiments. Naive CD4 + T cells were activated with plate-bound anti-CD3 (Clone: 145-2C11, 1 µg/ml, eBioscience), anti-CD28 (Clone: 37.51, 3 µg/ml, eBioscience), then polarized under Th1 condition with anti-IL-4 (10 µg/ml), IL-12 (10 µg/ml), IL-2 (100 U/ml).

Hepatocytes from Brg1-AID mice, Brg1^fl/fl Alb-Cre, and control mice were isolated by two-step collagenase perfusion and low-speed centrifugalizing following a procedure reported previously[47]. Briefly, after anesthetization of the mice, the portal vein was cannulated and firstly perfused with ethylene glycol tetraacetic acid (EGTA), followed by recirculation of 30 ml perfusion solution (0.075% Collagenase type I with 0.002% DNase I) at a flow rate 4 ml/min until hepatic parenchyma appear liquefied. The mouse livers were then digested in 15 ml digestion solution (0.009% Collagenase type I with 0.002% DNase I) for 10 min at 37 °C with rotation. To remove non-hepatic cells, the cell pellets were washed with 50 ml Hank's Balanced Salt Solution (HBSS) buffer for four times with centrifugation at $50 \times g$ for 5 min at 4 °C. Primary hepatocytes with > 95% purity and > 90% viability were considered for further experiments. Mouse primary hepatocytes were cultured in high glucose Dulbecco's Modified Eagle Medium (DMEM) medium containing 10% fetal bovine serum to maintain hepatocyte property[47]. To isolate hepatocytes from Brg1^fl/fl Alb-Cre mice and Brg1^fl/fl control littermates, mice were euthanized, and livers were immediately dissected, minced, and dissociated for 30 min in collagenase solution at 37 °C with rotation. Cell clumps were disrupted by pipetting and then washed and pelleted four times as above to obtain > 95% hepatocytes.

Primary fibroblasts were isolated from the ears of Brg1-AID mice and control mice. In brief, the ear was cut into 1 - 3 mm pieces, incubated with 0.25% trypsin–EDTA solution for 30 min at 37 °C, neutralized by culture medium with FBS and then plated on 6 well plate with coverslip on it. Fibroblasts will migrate from tissues during the 3 - 5-day culture period. After reaching the appropriate confluence, tissues will be removed and primary fibroblasts will be passaged and maintained in DMEM medium (high glucose) containing 10% fetal bovine serum.

Plasmids construction, hepatocytes and fibroblasts transfection, T cell transduction.

The osTIR1 expression vector was obtained from Addgene (Plasmid #86233)[48]. The retroviral osTIR1 expression vector (pSIREN-CMV-zsGREEN-CMV-osTIR1) was constructed by assembling the PCR-amplified CMV promoter sequence and osTIR1 DNA with the backbone of pSIREN-RetroQ-ZsGreen1 (Cat. No. 632455) using Gibson assembly cloning Kit following the manufacturer's protocols (NEB, E5510S). For transfection of hepatocytes, 2 µg plasmid DNA was used for $1 \times 10^6$ in vitro cultured hepatocytes using lipofectamine 2000 (Invitrogen, Cat. No. 11668-019) following the manufacturer's instruction. For transfection of fibroblasts, 20 µg plasmid DNA was used for each 10 cm dish using a combination of two commercial reagents, Lipofectamine LTX (Cat. No. 15338100) and FuGENE HD (Cat. No. E2311) with the ratio of 1:1[49]. The auxin was added 24 h later after transfection and incubated for 8 - 10 h. Then, the fibroblasts were treated with or without 1.0 µM Trichostatin A (TSA, SIGMA, T1952-200UL) for 2 h before sorting. The retroviral packaging and T-cell transduction were performed as previously described[36]. GFP+ cells were purified by flow cytometry sorting for further experiments.

## DNase-seq

DNase-seq assays were performed as described[30]. Briefly, 10,000 cells from each cell type were collected by FACS sorting. 0.3 unit of DNase I (Roche) was added to each cell type and incubated at 37 °C for 5 min. Reactions were stopped by adding 80 µl of stop buffer (10 mM Tris-HCl, 10 mM NaCl, 0.1% SDS, 10 mM EDTA) containing 1 µl of 20 mg/ml proteinase K. Samples were incubated at 65 °C overnight, and DNAs were purified by using MinElute reaction cleanup kit following the manufacturer's protocol. Purified DNA was end-repaired using an end-repair kit (ER81050, Epicenter), followed by A-addition using Klenow fragment (3' to 5' exo minus) and dATP. The Y-shaped adapters were then ligated to DNA ends using T4 DNA ligase. Libraries were amplified with Illumina primers for 16 cycles and library fragments from 150 bp to 300 bp were isolated for sequencing on Illumina HiSeq or NovaSeq platforms.

## MNase-seq

MNase-Seq assays were performed as described[50]. Briefly, 10,000 cells from each cell type were sorted by flow cytometer and deposited directly into 1.5 ml Eppendorf tubes, which contained 32 µl of cell lysis buffer (10 mM Tris-HCl, pH 7.5, 10 mM NaCl, 2 mM CaCl2, 0.1% Triton X-100). For MNase digestion, 8 µl diluted MNase (1:4000 in lysis buffer) was added to the cells and incubated at 37 °C for 5 min. The reaction was stopped by adding 80 µl stop buffer (10 mM Tris-HCl, pH 7.5, 10 mM NaCl, 0.15% SDS, 10 mM EGTA), and 2 µl of 20 mg/ml Proteinase K. Following incubation at 65 °C overnight, DNA was purified, libraries prepared and sequenced as described for the DNase-seq assays above.

## ChIC-seq and RNA-seq assays

ChIC-seq assays were performed as previously described[27,51]. Briefly, cells for ChIC-seq were fixed for 10 min with 1% formaldehyde and permeabilized with RIPA buffer (10 mM Tris-Cl, 150 mM NaCl, 0.1% SDS, 0.1% NaDOC, 1% Triton X-100) for 10 min at RT. 1 µl antibodies (anti-H3K4me3 (17-614, Millipore), anti-H3K27ac (ab4729, Abcam), anti-p300 (sc-584, Santa Cruz), anti-BRG1 (ab110641, Abcam), anti-RNA Polymerase II (ab817, Abcam)) and 3 µl protein A-MNase fusion protein were mixed well and incubated at 4 °C for 30 min. Next, the antibody-pA-MNase complex was added to the 100ul permeabilized cells and incubated at 4 °C for 1 h. Then, the cells were washed with 600 µl RIPA buffer four times and rinsed once with 200 µl rinsing buffer (10 mM Tris-Cl and 10 mM NaCl, 0.1% Triton X-100). The MNase digestion was initiated by re-suspending rinsed cells in 40 µl reaction solution buffer (10 mM Tris-Cl, 10 mM NaCl, 0.1% Triton X-100, 2 mM CaCl2) and incubated at 37 °C for 3 min. The reaction was stopped by adding 80 µl stop buffer (20 mM Tris-Cl, 10 mM EGTA, 20 mM NaCl, 0.2% SDS) and 1 µl 20 mg/ml proteinase K, then incubated at 65 °C overnight. DNA was purified by using a MinElute reaction cleanup kit following the manufacturer's protocol. The library preparation and sequencing were performed as described for the DNase-seq assay above.

RNA-seq was performed as described[52]. Briefly, total RNAs from 10,000 sorted cells were extracted using miRNAeasy Micro Kit (Qiagen, Cat#217084), followed by elution with 15 ul RNase-free water. Total RNAs from 2 K cells were reverse transcribed and pre-amplified by PCR using KAPA HiFi Hotstart Ready Mix (Roche Cat#KK2602) with IS PCR for 9 cycles. PCR products were purified by AMPure XP beads (Beckman Colter, Cat#A63881) and eluted with 17.5 µl nuclease-free water as described in Smart-seq2 method[53]. Amplified cDNAs (40 µl) were sonicated to 200–400 bp by Bioruptor Pico (Diagenode) for 20 cycles (30 s on and 30 s off). Library preparation using the sonicated cDNAs and sequencing were performed as described for DNase-seq.

## PRO-seq assay

PRO-seq (precision nuclear run-on sequencing) assays were performed as previously reported[28] with several steps modified as described below. Briefly, 500,000 CD4 + T cells were sorted by flow cytometry

from both Brg1-AID and control mice. Cells were permeabilized in 500 µl permeabilization buffer (10 mM Tris-HCl, pH 7.4, 300 mM sucrose, 10 mM KCl, 5 mM MgCl2, 1 mM EGTA, 0.05% Tween-20, 0.1% NP40 substitute, 0.5 mM DTT) for 5 min on ice with protease inhibitors and RNase inhibitor and then resuspended in 100 µL storage buffer (10 mM Tris-HCl, pH 8.0, 25% (vol/vol) glycerol, 5 mM MgCl2, 0.1 mM EDTA and 5 mM DTT). Then, the 100ul 2× nuclear run-on (NRO) master mix (10 mM Tris-HCl, pH 8.0, 5 mM $MgCl_2$, 300 mM KCl, 1 mM DTT, 50uM biotin-11-CTP, 0.5uM CTP, 0.25 mM ATP, 0.25 mM GTP, 0.25 mM UTP, RNase inhibitor, 1% Sarkosyl) were added into cells and incubated at 37 °C for 3 min. This step allows the engaged RNA polymerases to actively elongate and label the nascent RNA. Then, the nascent RNAs were extracted and hydrolyzed with NaOH to generate small RNA fragments, which could be enriched by affinity purification using M280 magnetic beads. Next, 3' adapters were ligated directly to the purified nascent RNA by incubate 10ul reaction mixture (1x T4 RNA ligase buffer, 1U/ul T4 RNA ligase I, 1 mM ATP, 10% PEG, 5uM 3' RNA adapter, RNase inhibitor) at 20 °C for 4 h. Following streptavidin purification, the 5' cap were removed from short nascent RNA using tobacco acid pyrophosphatase (TAP), then 5' ends were repaired using PNK and ligated to 5' adapters at 20 °C for 4 h in 10 µl mixture (1x T4 RNA ligase buffer, 1U/ul T4 RNA ligase I, 1 mM ATP, 10% PEG, 5uM 5' RNA adapter, RNase inhibitor). The adapter-containing RNA fragments are enriched by streptavidin pull-down, followed by reverse transcription and PCR amplification with Illumina primers which same as in the previous study for 18 cycles[28]. The fragments from 150 bp to 700 bp were isolated from E-gel and sequenced on Illumina NovaSeq. For fibroblasts and hepatocytes, 150,000 and 125,000 cells were sorted respectively and mixed with 5 million 293 T spike-in cells and then PRO-seq libraries were generated as above.

## Western blot analysis

300,000 CD4 + T cells, hepatocytes, and 40,000 fibroblasts lysed in 50 µl 2 X SDS loading buffer with 10% DTT were resolved by the Novex NuPage SDS-PAGE gel system (Life Technologies). Proteins were transferred to Supported Nitrocellulose Membrane (Bio-Rad) and were incubated with anti-Brg1 (sc-10768, Santa Cruz, 1:1000), anti-osTIR1(PD048, MBL life science, 1:1000), and anti-GAPDH (clone: AC-15, Sigma-Aldrich, 1:5000), anti-panH3 (ab21054, Abcam, 1:5000) and anti-actin (A2228, Sigma, 1:5000). Blots were visualized with Pierce ECL Western Blotting Substrate (Thermo Scientific).

## Data analysis

**Normalization.** Normalization using library size. Each set of reads was aligned to the mouse genome. The read count of each library was normalized by dividing it by the library size. This normalized count was then further scaled by a factor of $10^6$.

**Normalization using spike-in chromatin.** In our experimental methodology, which encompassed wild-type (WT) and various distinct conditions, individual sets of reads were aligned to both the mouse genome and the genome of the spike-in species. Consequently, reads that can be aligned to both reference genomes were filtered out. Each library's read count was then divided by a normalization factor, while also being scaled by a factor of $10^6$. The computation of this normalization factor involved two distinct components. The first factor was derived from the average number of mapped reads across all libraries, aligned to the mouse genome. The second factor was represented by the Spike-in normalization factors, calculated from the mapped reads aligned to the spike-in species.

For each specific condition, the count of reads mapped to the spike-in species was divided by the count of reads mapped to a selected library from the WT group. This intricate normalization procedure, encompassing multiple steps, assured precise and dependable quantification of the data. Consequently, this method enabled meaningful

comparisons between various experimental conditions, facilitating accurate interpretations and conclusions drawn from the data.

**Normalization using conserved genes.** This method was exclusively employed to investigate the nascent transcription quantified through PRO-seq analysis in T cells. The calculation of normalization factors for the libraries involved assessing the read counts linked to genes exhibiting robust expression (above a set threshold of 500 CPM) and the absence of Brg1 enrichment. Throughout this procedure, a meticulous selection process led to the inclusion of a total of 27 genes, serving the specific objective of determining these normalization factors.

**Removal of background noise in ChIC-seq and DNase-seq data.** In the analysis of ChIC-seq and DNase-seq data, a two-step normalization process was employed. Initially, the reads within peak regions were normalized based on the sizes of the respective peaks. Subsequently, the data underwent background noise correction. This correction involved estimating background noise through quantifying reads located outside peak regions, followed by division by the overall genome size. Further refinement was achieved by normalizing reads using the methods described in the subsection on normalization within the methods section.

**Statistical tests.** *Reproducibility-Optimized Test Statistic.* In this study, we utilized the Reproducibility-Optimized Test Statistic (ROTS)[54] to conduct differential analysis, with a specific emphasis on MA plots and volcano plots. Notably, ROTS has demonstrated superior performance when compared to Limma[55], edgeR[56], and DESeq[57] in detecting differential genes, particularly evident in bulk cell RNA-seq analyses involving spike-in RNA-seq data.

*Wilcoxon rank-sum test.* In this study, the p-value indicating the difference between two groups in the violin plot was calculated using the Wilcoxon rank-sum test.

*Hypergeometric test.* In this study, the p-value signifying the overlap between the two groups was determined through the Hypergeometric test.

**Analysis of Pro-seq data.** For CD4 + T cells, reads from PRO-seq were mapped to the mouse genome (mm10) using the Pro-seq pipeline[58]. As described in the sub-section of normalization above, normalization factors were calculated using conserved genes. For both Hepatocytes and Fibroblasts, reads from PRO-seq were mapped to the mouse genome (mm10) and the spike-in reference genome (hg19) using the Pro-seq pipeline. Libraries were normalized using spike-in normalizations. The differential genes in Pro-seq data as shown in Figs. 2c, 6e, f were determined using the *p*-value cutoff of 0.05 and fold-change cutoff of 1.3.

**Analysis of DNase-seq data.** For Th1 cells, Hepatocytes, and Fibroblasts, read data were mapped to the mouse genome (mm10) and the human spike-in genome (hg19) using Bowtie2. Libraries were normalized using spike-in normalizations. The differential sites were determined using the p-value cutoff of 0.05 and fold-change cutoff of 1.3. The enriched regions were identified using SICER[59] with suggested parameters.

**Analysis of Brg1, p300, H3K27ac, PolII ChIC-seq data.** For CD4 + T cells, ChIC-seq data were mapped to the mouse genome (mm10) using Bowtie2. Normalization factors for libraries were the total library sizes. For both Hepatocytes and Fibroblasts, ChIC-seq data were mapped to the mouse genome (mm10) and the spike-in human genome (hg19) using Bowtie2. Libraries were normalized using spike-in normalizations. The differential sites were determined using the p-value cutoff of 0.05 and fold-change cutoff of 1.3. The enriched regions were identified using SICER[59] with suggested parameters.

However, the enriched regions with the information of peak summit were identified using MACS2[60].

**Analysis of RNA-seq data.** For CD4 + T cells, reads from RNA-seq were mapped to the mouse genome (mm10) using Bowtie2. The normalized reads per kilobase per million mapped reads (RPKM) for each gene were calculated using HOMER[61]. Differential genes were determined using the p-value cutoff of 0.05 and fold-change cutoff of 1.5.

**Analysis of MNase-seq data.** Paired-end sequencing reads for MNase-seq were mapped to the mouse genome (mm10) using Bowtie2. Discordantly aligned reads, reads with mapping quality (MAPQ) ≤ 10, and redundant reads were removed from further analysis. We selected the fragments with a range of 140–180 bp as canonical nucleosomes for further analysis. The midpoints of nucleosomes were defined as their positions. We used a center-weighted occupancy score to plot aggregated nucleosome profiles at single base resolution. Specifically, we defined the nucleosome center positioning (NCP) score $S_k$ at position $k$, as the normalized nucleosome count with center at position $k$. We defined center-weighted occupancy score at position $k$ as $\sum_{j=-73}^{j=73} S_{k+j} w_j$, where $w_j$ is the Gaussian weight equal to $e^{-\left(\frac{j}{20}\right)^2 / 2}$. The height and location of peaks of the normalized nucleosome counts were identified using the 'findpeaks' function in Matlab, around both TSS and enhancers defined by the summit of p300 bindings. The MinPeakProminence was set to be 0.1. The two maximum peaks were recorded in the regions +/− 150 bp around the center. At both TSS and enhancers, the one-tailed z-score test was used to determine the shifting of + 1 and − 1 nucleosomes with a 90 percent confidence level. Three groups of shifting sites were identified.

### Reporting summary
Further information on research design is available in the Nature Portfolio Reporting Summary linked to this article.

### Data availability
The DNase-seq, MNase-seq, ChIC-seq, PRO-seq, and RNA-seq data generated in this study have been deposited in the GEO database under accession code GSE156569. The processed data of figures are available at FigShare [https://doi.org/10.6084/m9.figshare.24578932.v2]. Source data are provided with this paper.

### Code availability
Custom codes for generating figures are available at https://doi.org/10.6084/m9.figshare.24715956.

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

## Acknowledgements
We thank the National Heart, Lung, and Blood Institute Transgenic Core Facility for generating the AID knock-in mouse lines, the DNA Sequencing Core Facility for sequencing the libraries and the Flow Cytometry Core Facility for sorting the cells. The work was supported by Divisions of Intramural Research, National Heart, Lung and Blood Institute and National Institute of Environmental Health Sciences, Z01 ES071006-20.

## Author contributions
K.Z. conceived the project. G.R. and G.G. performed the experiments and W.L.K. performed the data analysis. K.C., J.A.H., J.Y.K., Y.H., Y.G., and Q.T. contributed to the experiments. C.L. led the NHLBI Transgenic Core facility for generating the AID knock-in mouse strains. B.G., T.K.A. and K.Z. directed the study. W.L.K, G.R, and K.Z. wrote the paper.

## Funding

## Competing interests
The authors declare no competing interests.
