## [Peer Review File · Nature Communications]

Acute depletion of BRG1 reveals its primary function as an activator of transcriptionEditorial Note: This manuscript has been previously reviewed at another journal that is not operating a transparent peer review scheme. This document only contains reviewer comments and rebuttal letters for versions considered at *Nature Communications*.

REVIEWER COMMENTS

Reviewer #2 (Remarks to the Author):

The authors have generally done a good revision by addressing my main initial concerns and I would recommend publication, should the minor points below be addressed.

Two minor points were not answered properly:

First, the description of the spike-in controls were only mentioned in the bio-informatics part concerning the ChIC-seq and DNase-seq and should be described for clarity in the protocol section (what amount of spike-in and how it was added). The analysis procedure used for spike-in normalization should either be described in more details or at least referenced.

Second, I am rather confident that RNA-seq data signal distribution is NOT normal, so again, the fisher test does not apply and should be replaced by appropriate testing. This detail is important because it could be misleading for younger scientist that could believe that normal tests such as t-test can be applied for any kind of data.

Reviewer #4 (Remarks to the Author):

In this revised manuscript from the Zhao group, the authors investigated the consequences of acute BRG1, an ATPase subunit of the chromatin remodeling SWI/SNF complex, deletion on chromatin accessibility and transcriptional regulation based an engineered AID-BRG1 knock-in mouse model, where experiments were performed on isolated T cells, hepatocytes, and fibroblasts. Through time-course BRG1 deletion, coupled with various assays such as ChIC-seq, PRO-seq, DNase-seq, they found that BRG1 mainly functions as transcriptional activator and directly regulates chromatin accessibility and RNA PolII targeting. Most of these findings are in agreement with several recent studies using acute BRG1 deletion or inhibition.

While the reviewer understood that this revision took over 2 years due to the pandemic, during this time, a few groups have published several related papers describing the direction activity of BRG1/BRM1 in regulating chromatin accessibility and transcription using similar approaches (targeted protein degradation or BRG1/BRM ATPase inhibitor and degraders, such as PMID: 34117481, 33558760, 33558759, 33558757), the novelty of this manuscript at its current form is somewhat reduced. It is also strange they did not acknowledge or cite any of these papers. While the authors have performed additional experiments such as adding new cell type (fibroblasts) during the revision, some major concerns (both conceptual and technical) raised by previous reviewers remain:

1. A major concern (raised by previous reviewer) is the degree of BRG1 degradation in these cells. It is understandable that different cell types have differential degradation efficiency, in Figure 1A, BRG1 protein level was reduced by somewhat 80-90% at 10h, yet in panel C, the decreased BRG1 binding signal is only ~20% of total peaks. The authors stated that there is a general trend of BRG1 loss, but ~80% of the peaks have less than 1.5 fold change, this seems contradicting the WB results. On a related note, in this system, the authors found that BRG1 peaks are almost equally enriched at TSS regions and enhancers/other regions. This is again, contradicting published results from many SWI/SNF labs showing the SWI/SNF binding is mostly at enhancers and only a small fraction (20-35%, depending on cell types/models, using ChIP-seq or CUT&RUN/ChIC-seq) is enriched at promoters. Could this be due to technical differences between ChIC-seq and CUT&RUN? Where the authors used fixation. Based on the genome browser examples the authors showed, while it's clear that there is reduced BRG1 signal in the AID group, the overall peak signal/intensity makes one wonder about the data quality. It would be helpful to show some additional analyses such as 1)

heatmaps of BRG1 signal: AID vs WT in addition to MA plot; 2) heatmap showing colocalization of BRG1 with major histone marks such K4me3, K4me1, and K27ac.

2. Another shared comment is about the compensating mechanism. Have the authors checked the level of BRM, the other APTase subunits, in these acute deletion experiments?

3. Since BRG1/BRM specific inhibitors (such as BRM014) or degraders (although less specific) are available, it would be helpful to add one of these inhibitors to compare AID-BRG1 degradation on some of the experiments such as DNase, PRO-seq. This would clarify some of the concerns raised by previous reviewers.

Some minor comments:

1. Lack of references to more recent publications and relevant papers and duplicated references such as Ref13, 14 and Ref 28, 29.
2. Methods section, especially the computational analysis parts still lack many details.
3. The mechanistic insight still feels thin.

We highly appreciate the reviewers' agreeing our previous revision. Their constructive and insightful feedbacks greatly contributed to the refinement of our manuscript. We have incorporated their valuable suggestions into our work and believe that we have successfully addressed all of the reviewers' concerns.

Reviewer #2:

The authors have generally done a good revision by addressing my main initial concerns and I would recommend publication, should the minor points below be addressed.

Two minor points were not answered properly:

First, the description of the spike-in controls were only mentioned in the bio-informatics part concerning the ChIC-seq and DNase-seq and should be described for clarity in the protocol section (what amount of spike-in and how it was added). The analysis procedure used for spike-in normalization should either be described in more details or at least referenced.

Second, I am rather confident that RNA-seq data signal distribution is NOT normal, so again, the fisher test does not apply and should be replaced by appropriate testing. This detail is important because it could be misleading for younger scientist that could believe that normal tests such as t-test can be applied for any kind of data.

Response: We are happy that the reviewer believes we have done good revision and has the main concerns. We also appreciate the remaining minor concerns of the reviewer on the clarity of the spike-in protocol and statistics test used in the analysis. To address these concerns, we have (1) clearly described our protocol for the spike-in control; and (2) employed a new method called "ROTS" [1] for analysis of differential gene expression or differential peak calling. The results from the new analysis strategy are consistent with our previous results and we updated the figures 1c, 2b, 2c, 2h, 3d, 4b, 4c, and 5a-f using these new results.

Reviewer #4:

In this revised manuscript from the Zhao group, the authors investigated the consequences of acute BRG1, an ATPase subunit of the chromatin remodeling SWI/SNF complex, deletion on chromatin accessibility and transcriptional regulation based an engineered AID-BRG1 knock-in mouse model, where experiments were performed on isolated T cells, hepatocytes, and fibroblasts. Through time-course BRG1 deletion, coupled with various assays such as ChIC-seq, PRO-seq, DNase-seq, they found that BRG1 mainly functions as transcriptional activator and directly regulates chromatin accessibility and RNA PolIII targeting. Most of these findings are in agreement with several recent studies using acute BRG1 deletion or inhibition.

While the reviewer understood that this revision took over 2 years due to the pandemic, during this time, a few groups have published several related papers describing the direction activity of BRG1/BRM1 in regulating chromatin accessibility and transcription using similar approaches (targeted protein degradation or BRG1/BRM ATPase inhibitor and degraders, such as PMID: 34117481, 33558760, 33558759, 33558757), the novelty of this manuscript at its current form is somewhat reduced. It is also strange they did not acknowledge or cite any of these papers. While the authors have performed additional experiments such as adding new cell type (fibroblasts) during the revision, some major concerns (both conceptual and technical) raised by previous reviewers remain:

1. A major concern (raised by previous reviewer) is the degree of BRG1 degradation in these cells. It is understandable that different cell types have differential degradation efficiency, in Figure 1A, BRG1 protein level was reduced by somewhat 80-90% at 10h, yet in panel C, the decreased BRG1 binding signal is only ~20% of total peaks. The authors stated that there is a general trend of BRG1 loss, but ~80% of the peaks have less than 1.5 fold change, this seems contradicting the WB results. On a related note, in this system, the authors found that BRG1 peaks are almost equally enriched at TSS regions and enhancers/other regions. This is again, contradicting published results from many SWI/SNF labs showing the SWI/SNF binding is mostly at enhancers and only a small fraction (20-35%, depending on cell types/models, using ChIP-seq or CUT&RUN/ChIC-seq) is enriched at promoters. Could this be due to technical differences between ChIC-seq and CUT&RUN?

Response: We thank the reviewer for the careful examination of our data. The reviewer is correct that the majority of BRG1 peaks identified in previous publications belonged to non-TSS regions. We checked the distribution of our BRG1 peaks from the ChIC-seq data and found that 46%, 34%, and 23% of BRG1 peaks from T cells, hepatocytes (**Figure R1**), and fibroblasts (**Figure R2**), respectively, are TSS peaks. Thus, our data are consistent with the literature reports, although there is some variation across different cell types.

Figure R1. A bar plot showing the number of the BRG1 peaks at TSS, enhancers, and other regions in the wild type Hepatocytes.

Figure R2. A bar plot showing the number of the BRG1 peaks at TSS, enhancers, and other regions in the wild type Fibroblasts.

- Where the authors used fixation. Based on the genome browser examples the authors showed, while it's clear that there is reduced BRG1 signal in the AID group, the overall peak signal/intensity makes one wonder about the data quality. It would be helpful to show some additional analyses such as 1) heatmaps of BRG1 signal: AID vs WT in addition to MA plot; 2) heatmap showing colocalization of BRG1 with major histone marks such K4me3, K4me1, and K27ac.

Response: We thank the reviewer for the suggestions of heatmap analysis. In response to the reviewer's suggestion, we have performed heatmap analyses that compare changes in BRG1, H3K4me3, and H3K27ac signals between WT and AID conditions. As indicated by the

heatmaps (Figures R3 and R4), there is a predominant reduction in H3K4me3 (Figure R3) and H3K27ac (Figure R4) density at the Brg1 peaks showing the most significant decrease in BRG1 signal.

Figure R3. The left panel of the figure displays heatmaps depicting the BRG1 fold-change between WT and AID conditions. The right panel illustrates the H3K4me3 fold-change between WT and AID states. These heatmaps are centered around TSS regions that co-localize with Brg1 peaks in either WT or AID CD4 T cells. In the color scale, higher fold-change values greater than 1 indicates increased density in the WT condition.

Figure R4. The left panel of the figure displays heatmaps depicting the BRG1 fold-change between WT and AID conditions. The right panel illustrates the H3K27ac fold-change between WT and AID states. These heatmaps are centered around enhancers regions (defined by H3K27ac) that co-localize with Brg1 peaks in either WT or AID CD4 T cells. In the color scale, higher fold-change values greater than 1 indicates increased density in the WT condition.

2. Another shared comment is about the compensating mechanism. Have the authors checked the level of BRM, the other APTase subunits, in these acute deletion experiments?

Response: To address the reviewer's question, we examined the gene expression levels of BRM (Smarca2) and 24 well established or potential subunits of the BAF complexes according to Anja F. Hohmann et al., (Trends in genetics, 2014). This analysis was carried out in *Brg1* fl/fl Th1 cells, comparing between CRE- and CRE+ conditions at both the 60-hour and 6-day time intervals. As illustrated in **Figure R5**, the gene expression of *Brg1* (Smarca4) exhibited a reduction at both time points.

Upon evaluating the remaining 24 subunits (refer to **Figures R5-R7**), we found that only Brm (Smarca2) consistently displayed increased expression across all replicates and time points in the CRE+ condition (**Figure R5**), suggesting that increased BRM expression may be a compensating mechanism for the loss of Brg1 in the cells.

Figure R5. The bar plots show the gene expression levels of nine subunits within the SWI/SNF complexes. A comparison is presented between Brg1 fl/fl Th1 CRE- (illustrated by blue bars) and CRE+ (represented by red bars), at both the 60-hour and 6-day time points. Notably, each time point includes three replicates.

Figure R6. Similar to Figure R5, showing additional nine subunits within the SWI/SNF complexes.

Figure R7. Similar to Figures R5 and R6, showing additional seven subunits within the SWI/SNF complexes.

3. Since BRG1/BRM specific inhibitors (such as BRM014) or degraders (although less specific) are available, it would be helpful to add one of these inhibitors to compare AID-BRG1 degradation on some of the experiments such as DNase, PRO-seq. This would clarify some of the concerns raised by previous reviewers.

Response: We thank the reviewer for this excellent suggestion. As suggested by reviewer, we treated CD4 T cells with or without BRM014 for 12 hrs and conducted PRO-seq and DNase-seq experiments. By comparing with untreated cells, we found that BRM014 treatment resulted in substantial decreases in PRO-seq and DNase-seq signals, as exemplified by the Genome Browser images (**Figures R8 and R9**). Using MA plots, we identified 1223 genes with significantly decreased PRO-seq signals while only 35 genes exhibited increased signals by the BRM014 treatment (**Figure R10**). Similarly, 5794 DHSs showed significantly decreased accessibility, while only 58 DHSs exhibited increased accessibility in the BRM014-treated cells (**Figure R11**). These results are consistent with our data obtained from the transient depletion of BRG1 using auxin treatment of AID-tagged Brg1 cells. Although BRM014 inhibits both BRG1 and BRM, the similarly chromatin accessibilities by auxin-induced BRG1 depletion or BRM014 inhibitor treatment suggest that it is BRG1 that plays a major role in maintaining the accessibility of chromatin in the cells. Thus, these results support our conclusions obtained from the auxin-induced depletion of BRG1 experiments.

Figure R8. A snapshot from the genome browser, showing Pro-seq data for both WT (represented by two replicates in blue) and BRM014 (indicated by two replicates in red).

Figure R9. A snapshot from the genome browser, showing DNase-seq data for both WT (represented by two replicates in blue) and BRM014 (indicated by two replicates in red).

Figure R10. An MA plot illustrated alterations in nascent transcripts within T cells, as quantified by PRO-seq following a 12-hour treatment with the BRM014 inhibitor. In the plot, blue dots denote genes that exhibit a statistically significant increase, while red dots indicate those with a significant decrease. The dashed line indicates genes showing no change in expression. The selection of significant differentially expressed genes was established using a threshold of $P < 0.05$ and a fold-change > 1.3 . The computation of P values was performed utilizing the ROTS method [1].

Figure R21. A volcano plot show changes in DNase-seq read density at DHSs following a 12-hour treatment with the BRM014 inhibitor. In this plot, blue dots represent DHSs exhibiting a statistically significant increase, while red dots indicate those displaying a significant decrease. The dashed line refers to DHSs without changes in DNase-seq density. Selection of DHSs with significant differences was

determined using a threshold of $P < 0.05$ and a fold-change > 1.3 . A P value was computed through the application of the ROTS method[1].

Some minor comments:

1. Lack of references to more recent publications and relevant papers and duplicated references such as Ref13, 14 and Ref 28, 29.

Response: We have included the recent publications mentioned by the reviewer and corrected the duplicated references.

2. Methods section, especially the computational analysis parts still lack many details.

Response: We have revised the methods to include more details.

3. The mechanistic insight still feels thin.

Response: We have included the increase in BRM expression as a potential compensatory mechanism for the loss of BRG1 in the cells.

References:

1. Suomi, T., et al., *ROTS: An R package for reproducibility-optimized statistical testing*. PLoS Comput Biol, 2017. **13**(5): p. e1005562.

REVIEWER COMMENTS

Reviewer #2 (Remarks to the Author):

The authors have addressed my previous concerns. The manuscript is my opinion now suitable for publication.

Reviewer #4 (Remarks to the Author):

The authors have performed some experiments and added some analysis to show that BRM014 treatment resulted in loss of chromatin accessibility but failed to go one step further to compare these BRM014 affected regions to the ones identified using BRG1 depletion. This analysis would help clarify the statement that " BRG1 plays a major role in maintaining the accessibility of chromatin in the cells." On the other hand, this would contradict the statement that "increased BRM expression may be a compensating mechanism for the loss of Brg1 in the cells". In addition, in FigR3, R4, the heatmap showing reduction of H3K4me3 and H3K27ac seems rather marginal if there is any reduction (a very small fraction?, it's unclear how many peaks are in these plots).

REVIEWER COMMENTS

Reviewer #2 (Remarks to the Author):

The authors have addressed my previous concerns. The manuscript is my opinion now suitable for publication.

Response: We thank the reviewer for the time spent on our manuscript, which helped to improve the manuscript.

Reviewer #4 (Remarks to the Author):

The authors have performed some experiments and added some analysis to show that BRM014 treatment resulted in loss of chromatin accessibility but failed to go one step further to compare these BRM014 affected regions to the ones identified using BRG1 depletion. This analysis would help clarify the statement that “BRG1 plays a major role in maintaining the accessibility of chromatin in the cells.” On the other hand, this would contradict the statement that “increased BRM expression may be a compensating mechanism for the loss of BRG1 in the cells”.

Response: We appreciate the reviewer’s comments and agree that a direct comparison between BRG1-depleted regions with the AID strategy and those affected by BRM014 treatment would strengthen our findings. To address this, we have generated a heatmap illustrating the DNase-seq read density at DHSs for both BRG1-depleted samples and samples treated with the BRM014 for 12 hours (**Figure R1**). The data suggest that the changes in chromatin accessibility are highly similar between BRG1 depletion and BRM014 treatment, which supports our statement that “BRG1 plays a major role in maintaining the accessibility of chromatin in the cells.”

Moreover, we have observed subtle differences between the two conditions, which may indicate that BRM could contribute to a compensatory mechanism for the loss of BRG1. However, it is important to note that the inhibitor experiment was conducted over a relatively short period, which might not be sufficient for the cells to initiate and complete compensation processes.

Figure R1. A heatmap show DNase-seq read density at DHSs for both BRG1-depleted samples and samples treated with BRM014 inhibitor for 12 hours. The DHSs are separated into three groups (decreased, increased, and unchanged) based on the changes of DNase-seq read density upon BRG1-depletion (AID versus WT).

In addition, in FigR3, R4, the heatmap showing reduction of H3K4me3 and H3K27ac seems rather marginal if there is any reduction (a very small fraction? it's unclear how many peaks are in these plots).

Response: Thank you for your comments regarding Figures R3 and R4, and for pointing out the need for clearer presentation of the heatmaps for H3K4me3 and H3K27ac reduction. We agree that the changes observed in the heatmaps may appear marginal. To give a clearer context, we provided the exact number of peaks presented in these heatmaps in the figure legends. Overall, we think that although the H3K4me3 and H3K27ac signals are compromised at some regulatory regions by depletion of BRG1, BRG1 mainly functions to maintain the chromatin accessibility at DHSs to facilitate transcriptional activation in the cells.

REVIEWERS' COMMENTS

Reviewer #4 (Remarks to the Author):

The authors have addressed my previous comments.